EMBO
Molecular Medicine

# A monocyte-centered framework for predicting immunochemotherapy efficacy in lung squamous cell carcinoma patients

Jiangnan Zhao[1,2,3,13], Zijin Wang [ID][4,5,13], Manyu Xiao [ID][4,5,13], Yeyuan Zhang[1,2,3], Boxin Liu [ID][4,5], Silin Chen[4,5], Yiping Tian[6], Dongqing Lv[7], Pingli Wang[8], Hai Song [ID][1,2,3], Yuefeng Wu[9], Jian Liu [ID][4,5,10,11,12 ✉] & Kai Wang [ID][1,2,3 ✉]

## Abstract

**Lung cancer is the leading cause of cancer-related mortality worldwide, with lung squamous cell carcinoma (LUSC) comprising 20–30% of cases. Immunochemotherapy (IC) is the standard first-line treatment for advanced LUSC, yet reliable predictors of therapeutic response remain unavailable. Using single-cell multi-omics profiling of paired pre- and post-treatment tumor and blood samples, we observed that patients responding to IC exhibited significantly higher baseline levels of peripheral blood monocytes, tumor-infiltrating classical monocytes, and APOBEC3A+ monocytes across both compartments compared with non-responders. These associations were independently validated in additional cohorts using routine complete blood count testing and multiplex immunofluorescence analysis of native tumor tissues. Our findings reveal monocyte-related parameters as clinically accessible indicators that link systemic immunity with the tumor microenvironment and hold promise for predicting IC responsiveness in patients with LUSC.**

**Keywords** APOBEC3A; Monocytes; Predictor; Biomarker; Immunochemotherapy
**Subject Categories** Cancer; Immunology; Respiratory System

## Introduction

Lung squamous cell carcinoma (LUSC) remains a major clinical challenge despite improvements in therapeutic strategies, with outcomes significantly inferior to those observed in lung adenocarcinoma (Herbst et al, 2020; Johnson et al, 2023). Accounting for 20–30% of lung cancer, LUSC is characterized by a paucity of targetable oncogenic drivers, thereby limiting the applicability and efficacy of targeted agents (Wang et al, 2025; Gettinger et al, 2021). Immune checkpoint inhibitor (ICI)-based immunochemotherapy (IC) is now the standard first-line treatment for advanced LUSC (Wang et al, 2025; Ricciuti et al, 2024), yet only a subset of patients derives durable benefit due to primary and acquired resistance (Wang et al, 2021). This highlights the urgent need for reliable predictors of IC response to support individualized treatment decision-making.

Previous biomarker research has largely centered on PD-L1 expression, tumor mutational burden (TMB), and tumor-infiltrating lymphocytes (Yu et al, 2019; Chan et al, 2019; Rakaee et al, 2023). However, their predictive performance remains inconsistent, partly due to spatial and temporal heterogeneity, methodological discrepancies, and dynamic immune remodeling during therapy (Herbst et al, 2016; Ono et al, 2017; Hong et al, 2020). Moreover, tumor-restricted signatures may not be robustly captured in peripheral blood, limiting their utility for minimally invasive prediction (Tsai et al, 2024). Thus, tractable indicators capable of accurately reflecting IC responsiveness remain an unmet requirement in clinical oncology.

Monocytes and their subsets, including classical monocytes, have emerged as key orchestrators of tumor–immune interactions through differentiation into myeloid cell lineages and modulation of the tumor

[1]Department of Respiratory and Critical Care Medicine, the Fourth Affiliated Hospital of School of Medicine, and International School of Medicine, International Institutes of Medicine, Zhejiang University, Yiwu 322000, China. [2]Zhejiang Key Laboratory of Precision Diagnosis and Treatment for Lung Cancer, Yiwu 322000, China. [3]Zhejiang-Sweden Joint Laboratory on Tumor Immunology, Yiwu 322000, China. [4]Centre for Infection Immunity and Cancer (IIC) of Zhejiang University-University of Edinburgh Institute (ZJU-UoE Institute), Department of Respiratory and Critical Care Medicine, the Second Affiliated Hospital, Zhejiang University School of Medicine, Zhejiang University, Hangzhou 310029, China. [5]Edinburgh Medical School: Biomedical Sciences, College of Medicine and Veterinary Medicine, The University of Edinburgh, Edinburgh, UK. [6]Department of Pathology, Zhejiang Cancer Hospital, Hangzhou Institute of Medicine (HIM), Chinese Academy of Sciences, Hangzhou 310022, China. [7]Department of Respiratory and Critical Care Medicine, Taizhou Hospital of Zhejiang Province of Wenzhou Medical University, Taizhou 317000, China. [8]Department of Respiratory and Critical Care Medicine, the Second Affiliated Hospital of Zhejiang University School of Medicine, Hangzhou 310052, China. [9]Westlake Laboratory of Life Sciences and Biomedicine, School of Life Sciences, Westlake University, Hangzhou, Zhejiang 310030, China. [10]Zhejiang Key Laboratory of Medical Imaging Artificial Intelligence, Haining, China. [11]Biomedical and Health Translational Research Center of Zhejiang Province, Haining, China. [12]Key Laboratory of Cancer Prevention and Intervention, China National Ministry of Education, The Second Affiliated Hospital, Zhejiang University School of Medicine, Hangzhou, China. [13]These authors contributed equally: Jiangnan Zhao, Zijin Wang, Manyu Xiao. ✉E-mail: JianL@intl.zju.edu.cn; kaiw@zju.edu.cn

microenvironment (Chen et al, 2023; Olingy et al, 2019). Several recent studies suggest that monocyte-based metrics may possess prognostic or predictive value across cancer types (Levy et al, 2025; Carroll et al, 2023; Shao et al, 2021; Ezdoglian et al, 2025; Sekine et al, 2018; Kwiecień et al, 2020; Zilionis et al, 2019). Ratios such as monocyte-to-lymphocyte (MLR) and lymphocyte-to-monocyte (LMR) have also been evaluated in patients receiving IC (Shao et al, 2021; Michailidou et al, 2021; Zheng et al, 2023; Goldschmidt et al, 2023; Bisschop et al, 2019; Chen et al, 2022; Ribas et al, 2016). However, the predictive relevance of monocyte subset composition—particularly its spatial distribution between blood and tumors—and functional states in LUSC treated with IC remains poorly understood (Chen et al, 2023; Ezdoglian et al, 2025). A systematic assessment of monocyte dynamics during therapy may therefore provide mechanistic and predictive insights.

Apolipoprotein B mRNA Editing Enzyme Catalytic Subunit 3A (APOBEC3A), a cytidine deaminase contributing to genomic instability and a defined APOBEC mutational signature, is associated with elevated TMB and enhanced antitumor immune activity (Petljak et al, 2022; Jalili et al, 2020; Gupta et al, 2025; Yang et al, 2024b). While APOBEC has been proposed as a potential immunotherapy-related biomarker, its predictive significance in LUSC under IC has yet to be elucidated (Hata and Larijani, 2024). Understanding this relationship may enable refined stratification of treatment sensitivity versus resistance.

In this study, we identified circulating native monocyte ratio, native tumor classical monocyte ratio, and APOBEC3A+ monocytes in both native peripheral blood and tumors as predictive indicators of pathologic complete or major responses (pCR/MPR) to IC in LUSC. These findings established monocyte-derived features as informative indicators linking systemic and intratumoral immunity, and provided a rationale for incorporating APOBEC3A-related features into predictive frameworks. Ultimately, this approach offered a minimally invasive strategy to enhance response prediction, treatment selection, and clinical outcomes in patients with LUSC.

## Results

### Native peripheral blood monocyte ratio was significantly higher in IC-response LUSC patients than in non-responders

To identify potential predictors of IC response in patients with LUSC, we collected 23 peripheral blood (Native $n = 9$, Treated $n = 14$) and 26 primary tumors (Native $n = 12$, Treated $n = 14$) samples from 15 LUSC patients and performed paired single-cell RNA sequencing (scRNA-seq) and TCR sequencing. Patients were stratified into response or non-response groups according to predefined criteria (Fig. 1A; Table EV1) (Hellmann et al, 2014; Cottrell et al, 2018). We performed multivariate analyses comparing these variations between pCR/MPR and non-MPR (NMPR) patients and found no significant difference in their distribution (Fig. EV1A; Table EV2). Data preprocessing included quality control, removing doublets (Table EV3), batch-effect correction, and dimensionality reduction (Fig. 1B, see Methods). Dimensionality reduction prior to correction revealed a great batch of effects in monocytes and tumor cells (Fig. EV1B). Major

cell types were annotated based on established markers from previous non-small cell lung cancer (NSCLC) studies, and the annotations of tumor cells were confirmed through copy number variation (CNV) analysis (Figs. 1C,D and EV1C,D) (Salcher et al, 2022). No circulating tumor cells were detected in peripheral blood samples (Fig. EV1D). The annotation of tumor cells was also validated by Immunofluorescence (IF) in tumor sections of patients in the scRNA-seq cohort (Fig. EV1E). Given the substantial variability in neutrophil proportion (Fig. EV1F), we excluded neutrophils from downstream analyses as we aimed to identify stable marker candidates for predicting IC efficacy. The annotation of cell types in the native primary tumor microenvironment was also validated by proportion comparison with published LUSC tumor scRNA-seq data (Fig. EV1G) (Salcher et al, 2022). Quantitative comparisons of cell proportions revealed that only peripheral blood monocyte proportions differed significantly between pCR/MPR and NMPR patients before treatment (Fig. 1E,F; Appendix Fig. S1A). In contrast, no cell types in primary tumors differed between the groups (Fig. 1E,F; Appendix Fig. S1A). Post-treatment mast cell proportions in primary tumors differed significantly between pCR/MPR and NMPR groups (Fig. 1E; Appendix Fig. S1A). Additionally, NMPR patients exhibited more substantial post-treatment shifts in the tumor microenvironment composition than pCR/MPR patients, whereas no comparable changes were observed in peripheral blood (Fig. 1E; Appendix Fig. S1A). To validate the positive relationship between the native blood monocyte ratio and the responsiveness of LUSC patients, we collected complete blood count (CBC) test data from LUSC patients before treatment ($n = 228$, Table EV4). Compared to the discovery cohort mainly from 2023 (Table EV1), we collected more native CBC data of patients from 2024 to strengthen the translational credibility (Fig. 1G; Table EV4). The monocyte CBC ratio (excluding granulocytes) of pCR/MPR patients was significantly higher than that of NMPR patients before treatment (Fig. 1H). These findings suggested a positive association between native peripheral blood monocyte proportions and IC responses in patients with LUSC.

To further investigate transcriptomic differences in monocytes of peripheral blood between pCR/MPR and NMPR patients before treatment, we identified differentially expressed genes (DEGs) in monocytes from these two groups. Subsequently, we conducted the Gene Set Enrichment Analysis (GSEA) and found that monocytes in pre-treatment pCR/MPR blood had higher enrichment in pathways about responses to bacteria (Appendix Fig. S1B,C). Cell–cell communication analysis indicated that the interaction strength between monocytes and most other cells was weaker in native pCR/MPR peripheral blood than in NMPR samples (Fig. EV2A). In contrast, the interaction strength between monocytes and most other cells was stronger in pre-treatment pCR/MPR primary tumors compared to NMPR tumors (Fig. EV2A). Notably, these observed phenotypes were partially reversed after IC (Fig. EV2A). Similar to monocytes, interactions among other cell types exhibited comparable patterns before and after IC (Fig. EV2B). In short, these results suggested that pre-treatment peripheral blood monocytes in pCR/MPR LUSC patients displayed weaker interactions with other cells (e.g., T cells) than those in NMPR LUSC patients.

To investigate T cell dynamics and the relationship between T cells and IC response, we analyzed TCR clonal diversities and clone subtypes. There were no significant differences observed in overall TCR diversities between the pCR/MPR and NMPR patients

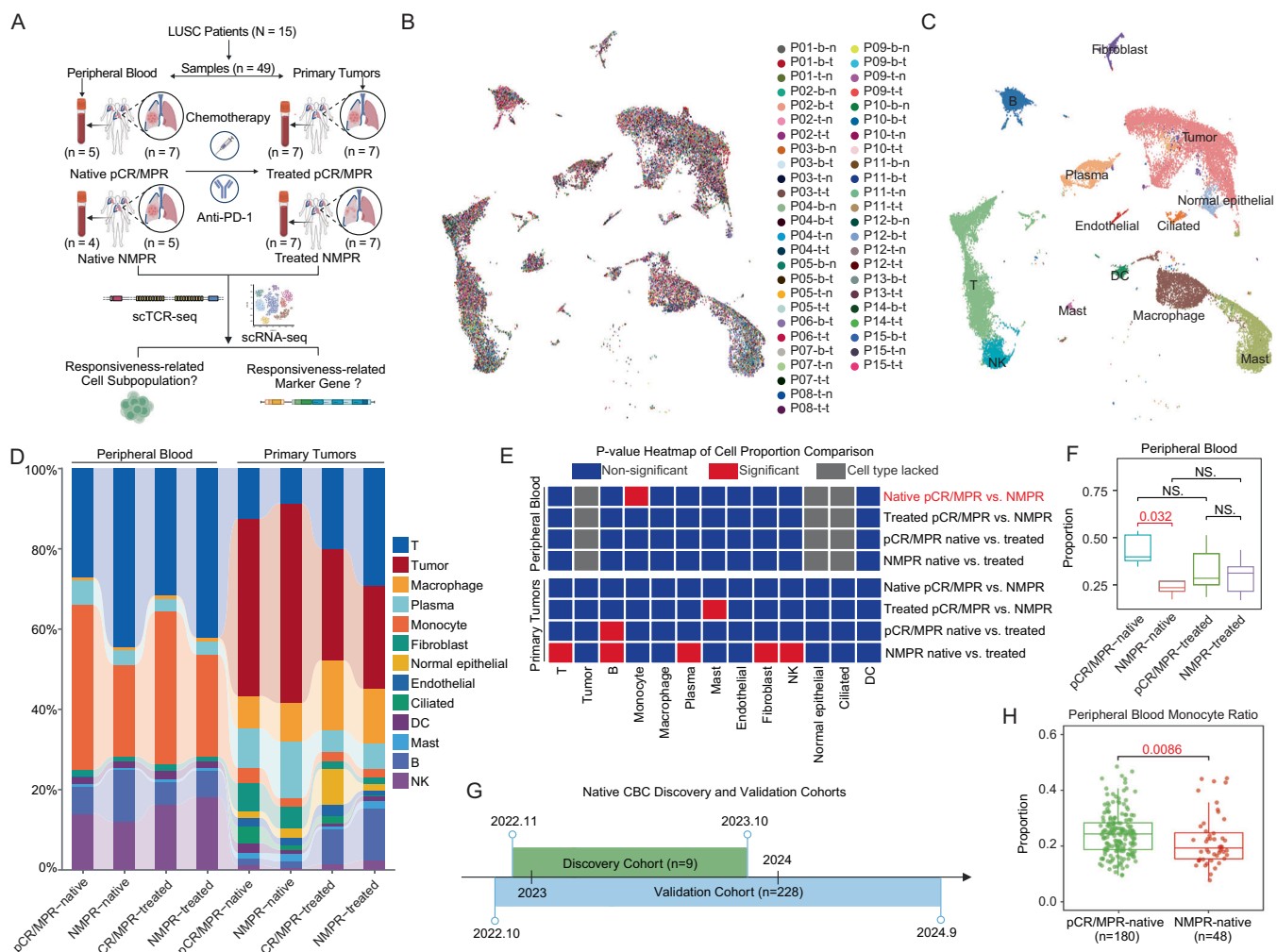

**Figure 1. Native peripheral blood monocyte proportions positively correlate with IC response in patients with LUSC.**

(A) Schematic diagram of sampling and sequencing strategy. (B) UMAP visualization of integrated cell data from peripheral blood and tumors ($n = 489,128$ cells). Sample IDs indicate patient ID (PXX), sample source (b: blood, t: tumor), and treatment status (n: native, t: treated). (C) UMAP embedding colored based on major annotated cell types. (D) Cell-type proportions in blood and tumor samples stratified using patient response groups before and after treatment. (E) Heatmap showing p-values from comparisons of cell-type proportions across different response groups in blood and tumors pre- and after-treatment (red color indicates significant differences, blue color indicates no significance, and grey means no enough cell numbers to compare; two-sided Wilcoxon test, significant $p < 0.05$). (F) Box plot and comparisons of monocyte proportions across response groups and between pre- and after-treatment (red color indicates significant differences; two-sided Wilcoxon test, significant $p < 0.05$, NS: no significance; blood sample $n_{blood} = 23$, before treatment $n_{pCR/MPR} = 5$, $n_{NMPR} = 4$, after treatment $n_{pCR/MPR} = 7$, $n_{NMPR} = 7$). Box plots followed the Tukey style (Centre line: median. Box bounds: 25th and 75th percentiles. Whiskers: extending to the most extreme data points within 1.5 times the interquartile range (IQR) from the box bounds. Outliers: points beyond the end of the whiskers). (G) Collection time distribution of the peripheral blood discovery and validation cohorts. (H) Box plot and comparisons of monocyte complete blood count (CBC) ratios (without granulocytes) between pCR/MPR and NMPR patients (pCR/MPR $n = 180$, NMPR $n = 48$). Comparison was conducted using two-sided Wilcoxon test (significant $p < 0.05$). Box plots followed the Tukey style (Centre line: median. Box bounds: 25th and 75th percentiles. Whiskers: extending to the most extreme data points within 1.5 times the interquartile range (IQR) from the box bounds. Outliers: points beyond the end of the whiskers). Source data are available online for this figure.

(Fig. EV2C). Hyperexpanded TCR clones were absent in native pCR/MPR blood, but were observed in a subset of treated pCR/MPR blood (Fig. EV2D). Hyperexpanded TCR clones were in a subset of native NMPR blood, and more treated NMPR blood had hyperexpanded TCR clones (Fig. EV2D). Conversely, no hyperexpanded TCR clone was found in native tumors, and only partially treated NMPR tumors had hyperexpanded TCR clones. This suggested that IC induced hyperexpanded TCR in blood and tumors of partial LUSC patients. In conclusion, compared with NMPR LUSC patients, these results suggested that peripheral blood

monocytes showed a higher ratio and weaker interactions with other cells (e.g., T cells) in native pCR/MPR LUSC patients, some of whom had native blood T cells without hyperexpanded TCR clones.

## Classical monocyte proportions in native LUSC tumors in the IC-response group were higher than those in the non-response group

To identify specific IC response predictors from the zoom-in analysis of cell subtypes in tumors, we annotated and compared

subpopulations of monocytes, macrophages, B cells (including plasma cells), T cells, and fibroblasts among different response groups (Fig. 2A–C; Appendix Figs. S2A and S3A). Subpopulations of these major cell types were annotated using established markers from previous studies (Zhang et al, 2018; Yang et al, 2024a; Lv et al, 2017; Yang et al, 2025; Delaloy et al, 2022; Hao et al, 2022). Monocyte was the only cell type whose classic subpopulation proportion in native tumors significantly differed between pCR/MPR and NMPR patients (Fig. 2C; Appendix Figs. S2A and S3A), with higher levels in native pCR/MPR tumors than in NMPR tumors (Appendix Fig. S2A). To validate this result, we collected pre-treatment primary tumor samples for IF staining (Table EV5). Given that the discovery cohort was primarily from 2023 (Table EV1), we augmented it with native tumor samples collected in 2024 to strengthen the translational robustness of our results (Fig. 2D; Table EV6). We also performed a priori sample-size assessment for the key independent tumor-tissue validation cohort used to evaluate candidate tumor-derived indicators for predicting pCR/MPR (detailed in Methods). IF staining of pre-treatment tumors confirmed a significant decrease in tumor-infiltrated classical monocytes in NMPR patients, corresponding to the increased residual viable tumor (RVT) (Fig. 2E, quantification detailed in Methods). In summary, we found that classical monocyte proportions in native tumors might be positively correlated with IC response.

To further investigate the differences of classical monocytes in primary tumors between pCR/MPR and NMPR patients before treatment, we identified DEGs (e.g., *APOBEC3A*, *IFI6*) and conducted GSEA. Compared to NMPR, classical monocytes in native pCR/MPR tumors showed enrichment in pathways such as responses to the virus, responses to cytokine, cell surface receptor signaling, cellular response to cytokine stimulus, and viral genome replication (Fig. EV3A,B).

Similar to the interaction patterns observed in monocytes (Fig. EV2A), classical monocytes from pCR/MPR patients exhibited weaker interactions with most other cell types (e.g., T cells) in native blood compared to NMPR patients (Fig. EV3C). Conversely, within native tumor samples, these results suggested that these interactions were stronger in pCR/MPR patients than in NMPR patients (Fig. EV3C).

IC treatment enhanced the interaction strength of classical and intermediate monocytes as receptors with most cell types—including intermediate monocytes, T cells, macrophages, DC cells, mast cells, and NK cells—in blood samples of pCR/MPR patients compared to NMPR patients (Fig. EV3C,D). In contrast, within tumors, IC treatment reduced the interaction strength of classical and intermediate monocytes as receptors with various cell types—including all three monocyte subtypes, macrophages, fibroblasts, normal epithelial cells, endothelial cells, DC cells, mast cells, B cells, and NK cells—in pCR/MPR compared to NMPR patients (Fig. EV3C,E). In summary, classical monocytes in native tumors from pCR/MPR patients showed potentially stronger interactions with other cell types (e.g., T cells) than those in tumors from NMPR patients, except with tumor cells, plasma cells, and ciliated cells.

To investigate the dynamics of T cell subtypes and the relationship between T cells and IC response, we analyzed TCR clone subtypes across T cell subpopulations. The percentage of large clones in certain native tumors of NMPR was higher than that in pCR/MPR patients (Fig. EV3F). Furthermore, CD8 + T cells (CD8 + $T_{EMRA}$/$T_{EFF}$: effector memory or effector T cells, CD8 + $T_{CM}$: CD8+ central memory T cells, and CD8 + $T_{EX}$: exhausted CD8$^+$ T cells) primarily contributed to the previously identified hyperexpanded clones observed in blood and tumor samples (Fig. EV3F). Similarly, the proportion of hyperexpanded clones in native blood samples of a subset of NMPR patients was higher than in pCR/MPR patients (Fig. EV3F). IC treatment increased the percentage of hyperexpanded clones in both blood and tumor samples, except in tumors from pCR/MPR patients (Fig. EV3F). In summary, these results suggested that native tumors of pCR/MPR patients exhibited fewer large TCR clones and no hyperexpanded TCR clones, yet demonstrated a higher ratio and potentially stronger interactions of classical monocytes with other cell types (e.g., T cells) compared to NMPR patients.

## APOBEC3A+ monocytes in native peripheral blood and tumors in IC-responding LUSC patients were higher than those in non-responders

To identify conserved IC-response predictors across blood and tumor tissues, we conducted a series of analyses combining blood and tumor samples, including DEG overlap analyses. To ensure the precision of selected IC-response predictors, we evaluated potential outliers and influential samples by imitating case-deletion diagnostics (CDD) (Figs. EV4A–C) (Christensen et al, 1992; De Bastiani et al, 2018). Consequently, we removed one patient sample (#P12-b-n) before predictor identification because it exerted an outlier effect on DEGs among response groups in native blood (Figs. EV4A–C). Then, we performed an overlap analysis of DEGs (pCR/MPR monocytes versus NMPR monocytes) between native blood and tumors (Fig. 3A,B). We identified *APOBEC3A* as the most consistently overexpressed gene in monocytes from peripheral blood and tumor tissues of pCR/MPR patients before treatment (Fig. 3A–C). Uniform manifold approximation and projection for dimension reduction (UMAP) visualization indicated subpopulation-specific *APOBEC3A* expression in monocyte subtypes of blood and tumors (Fig. 3C). APOBEC3A+ monocytes showed relative enrichment in non-classic and intermediate monocytes in the native blood of pCR/MPR patients (Figs. 2A and 3C). Meanwhile, APOBEC3A+ monocytes represented a shared population among these three subtypes in native tumors of pCR/MPR patients (Figs. 2A and 3C). There were significantly higher proportions of APOBEC3A+ monocytes in pCR/MPR native blood and tumors than in NMPR (Fig. 3D). These significant differences were not observed after IC (Fig. 3D), suggesting that IC altered the presence of APOBEC3A+ monocytes in both blood and tumors. In brief, *APOBEC3A* expression in monocytes in pCR/MPR native peripheral blood and tumors was higher than in NMPR.

IF staining of pre-treatment tumors confirmed a progressive decrease in APOBEC3A expression of tumor-infiltrated monocytes, corresponding to increased RVT (Fig. 3E). Quantitative analysis confirmed that the ratio of APOBEC3A+ monocytes versus all monocytes in pCR/MPR patients' native tumors was significantly higher than that of NMPR patients (Fig. 3E, detailed in Methods). Quantitative analysis also showed that approximately 70% of APOBEC3A+ cells were monocytes in the native tumors, indicating enriched APOBEC3A expression in tumor monocytes (Fig. EV4D). Therefore, we directly compared the APOBEC3A+

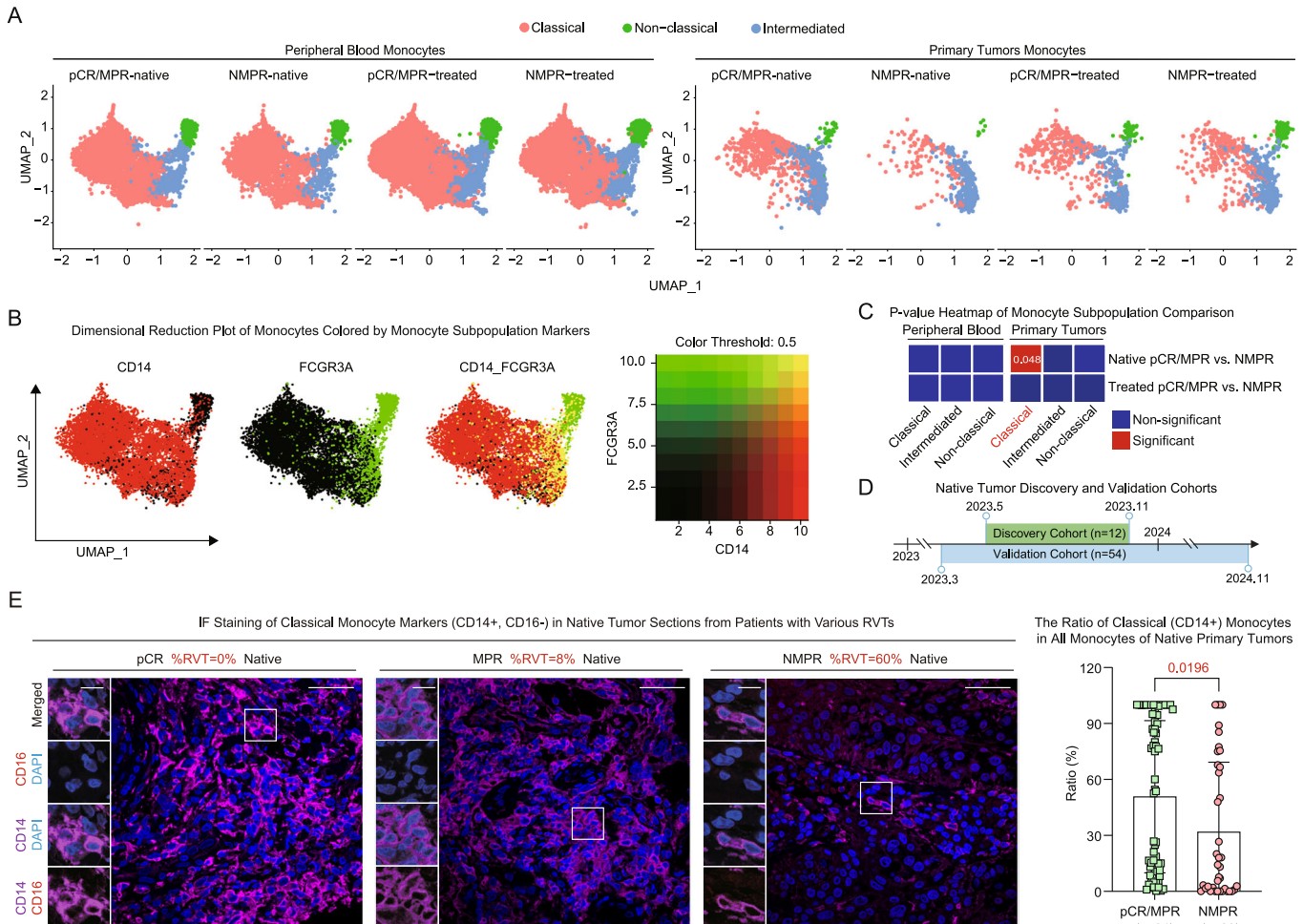

**Figure 2. Classical monocyte proportions in native tumors positively correlate with IC response.**

(A) UMAP visualization of monocyte populations colored using previously published subpopulation annotations ($n = 40,431$ cells). (B) UMAP embedding illustrating co-expression density markers defining classical monocytes (*CD14*), non-classical monocytes (*FCGR3A*), and intermediate monocytes (*CD14* and *FCGR3A*). (C) Heatmap depicting *p*-values from comparisons of monocyte subpopulation proportions among different response groups (red color indicates significant differences; two-sided Wilcoxon test, significant $p < 0.05$). (D) Collection time distribution of the primary tumor discovery and validation cohorts. (E) Multiplex IF staining of CD14 (purple), CD16 (red), and DAPI (blue) in pre-treatment LUSC tumor sections from patients with different responsiveness; scale bars: zoomed-in 10 μm; overview 40 μm; Quantity of classical monocytes in pre-treatment LUSC tumor sections from patients with different responsiveness (two-sided T-test, significant $p < 0.05$; pCR/MPR $n = 34$, NMPR $n = 20$). Error bars of bar plots represented the standard deviation. Source data are available online for this figure.

cell ratio and found that APOBEC3A+ cells in pCR/MPR native tumors were significantly higher than in NMPR tumors, suggesting that APOBEC3A expression alone can help predict LUSC patients' responsiveness without the need for monocyte markers (Fig. 3E). In summary, APOBEC3A positive expression in monocytes was identified and validated as a positive predictor for IC response in peripheral blood and tumors.

To enhance the clinical utility of the three identified indicators, we designed an application strategy that balances prediction accuracy and coverage by iteratively testing combinations and thresholds (Fig. EV4E; Table EV9, detailed in Methods). The combination of indicators and thresholds was calculated from scRNA-seq data and subsequently validated using paired CBC test data and IF staining results from native tumor slices (pCR/MPR $n = 22$, NMPR $n = 11$). The pCR/MPR coverage was defined as the ratio of the number of pCR/MPR patients who passed thresholds to

the total number of pCR/MPR patients ($n = 22$). The pCR/MPR accuracy was defined as the ratio of the number of pCR/MPR patients passing thresholds versus the sum of patients passing thresholds. In summary, we identified three optimal indicator combinations: the CBC monocyte ratio alone, which provided the broadest coverage and highest accuracy among single indicators; a two-indicator panel combining either the CBC or CD14+ monocyte ratio with the tumor APOBEC3A+ monocyte ratio, which achieved better accuracy; and the combination of all three indicators, which also yielded promising accuracy (Table EV9).

The interactions between APOBEC3A+ monocytes and other cells were weaker in pCR/MPR patients' native blood than in NMPR (Fig. EV5A,B). At the same time, they were stronger across most cell types in native tumors (Fig. EV5A), including APO-BEC3A- monocytes, T cells, macrophages, fibroblasts, normal epithelial cells, endothelial cells, DCs, Mast cells, and B cells. This

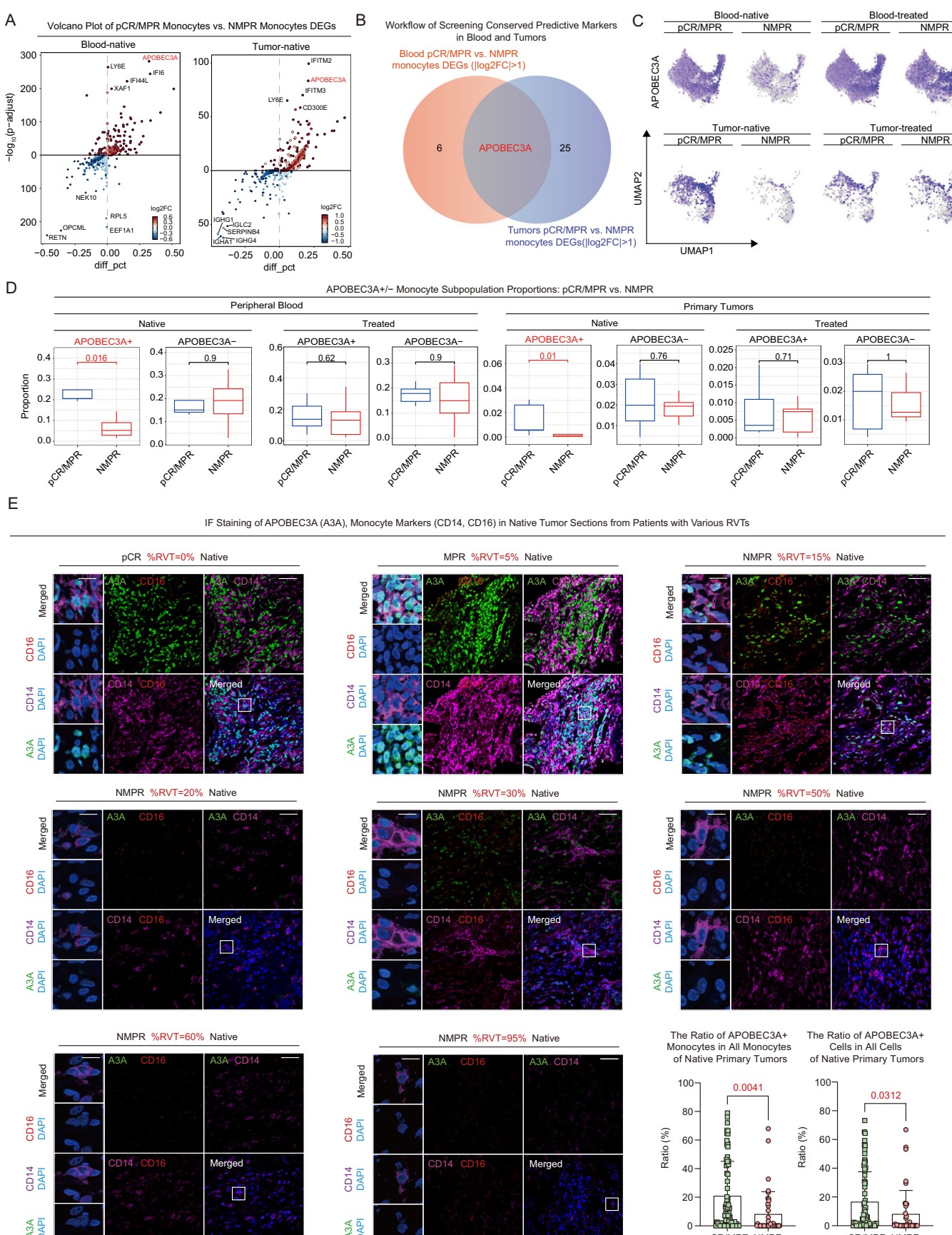

**Figure (A–E)**

A. Volcano Plot of pCR/MPR Monocytes vs. NMPR Monocytes DEGs (Blood-native, Tumor-native)

B. Workflow of Screening Conserved Predictive Markers in Blood and Tumors

C. APOBEC3A UMAP plots (Blood-native, Blood-treated, Tumor-native, Tumor-treated; pCR/MPR, NMPR)

D. APOBEC3A+/− Monocyte Subpopulation Proportions: pCR/MPR vs. NMPR

E. IF Staining of APOBEC3A (A3A), Monocyte Markers (CD14, CD16) in Native Tumor Sections from Patients with Various RVTs

◀ **Figure 3. APOBEC3A+ monocytes in native peripheral blood and tumors predict better IC treatment response.**

(A) Volcano plot of monocyte DEGs across response groups of native blood (blood sample $n_{blood} = 8$, before treatment $n_{pCR/MPR} = 5$, $n_{NMPR} = 3$) and primary tumors (tumor sample $n_{tumor} = 12$, before treatment $n_{pCR/MPR} = 7$, $n_{NMPR} = 5$) after CDD; the x-axis indicates percentage differences in gene-expressing cells between groups, and the y-axis shows $-\log_{10}$(adjusted $p$-value). Dots are colored based on $\log_2$FC values. Two-sided Wilcoxon test was utilized to calculate $p$-values and $p$-values were adjusted by Bonferroni correction. (B) Workflow showing conserved predictive markers in blood and tumors, defined by overlapping blood and tumors' pCR/MPR vs. NMPR monocytes DEGs with absolute $\log_2$FC > 1 ($|\log_2$FC $| > 1$) before treatment. (C) UMAP embedding of monocytes colored by *APOBEC3A* expression. (D) Box plot and comparisons of APOBEC3A+/− monocyte proportions across response groups pre- and after-treatment (two-sided Wilcoxon test, significant $p < 0.05$; blood sample $n_{blood} = 22$, before treatment $n_{pCR/MPR} = 5$, $n_{NMPR} = 3$, after treatment $n_{pCR/MPR} = 7$, $n_{NMPR} = 7$; tumor sample $n_{tumor} = 26$, before treatment $n_{pCR/MPR} = 7$, $n_{NMPR} = 5$, after treatment $n_{pCR/MPR} = 7$, $n_{NMPR} = 7$). Box plots followed the Tukey style (Centre line: median. Box bounds: 25th and 75th percentiles. Whiskers: extending to the most extreme data points within 1.5 times the interquartile range (IQR) from the box bounds. Outliers: points beyond the end of the whiskers). (E) Multiplex IF staining of CD14 (purple), CD16 (red), and APOBEC3A (green) in pre-treatment LUSC tumor sections from patients with varying residual tumor loads; scale bars: zoomed-in 10 μm; overview 40 μm; Comparison of APOBEC3A+ monocyte ratios in native pCR/MPR and NMPR tumor section; two-sided t-test (**$p < 0.01$; pCR/MPR $n = 34$, NMPR $n = 20$); Comparison of APOBEC3A+ cell ratios in native pCR/MPR and NMPR tumor section; two-sided t-test (**$p < 0.01$; pCR/MPR $n = 34$, NMPR $n = 20$). Error bars of bar plots represented the standard deviation. Source data are available online for this figure.

demonstrated distinct cell–cell interaction patterns of APOBEC3A + monocytes between blood and tumors (Fig. EV5A). In summary, these results suggested that APOBEC3A+ monocytes had cell–cell interaction patterns similar to those of all monocytes in native blood and classical monocytes in native primary tumors between pCR/MPR and NMPR patients, except for the interaction with APOBEC3A+ monocytes themselves (Figs. EV2A,EV3C, and EV5A).

Pathway enrichment analysis revealed highly conserved, significantly upregulated pathways related to antiviral defense mechanisms, positive regulation of cytokine production, and regulation of the response to biotic stimuli in APOBEC3A+ monocytes from blood and tumors (Fig. EV5C). Interestingly, the downregulated pathways were inconsistent between blood and tumors (Fig. EV5C). In brief, APOBEC3A+ monocytes consistently enriched genes related to antiviral and immune response pathways in both blood and tumors.

In summary, by conducting single-cell multi-omics analyses of paired pre- and post-treatment human LUSC tumor and blood samples, we observed that patients responding to IC showed significantly higher levels of native peripheral blood monocytes, native tumor-infiltrating classical monocytes, and native APOBEC3A+ monocytes across both compartments compared with non-responders. These associations were independently validated in additional cohorts using routine complete blood count testing and multiplex immunofluorescence analysis of native tumor tissues. Therefore, we identified a monocyte-centered framework for predicting IC efficacy in the LUSC patients (Fig. 4).

## Discussion

For advanced LUSC patients, immunochemotherapy represents the first-line treatment, yet reliable biomarkers for predicting its efficacy are extremely limited. Crucially, conducting multi-omics analyses on matched pre- and post-treatment samples from the same patient is quite challenging. To our knowledge, no prior study has performed matched single-cell multi-omics profiling on both peripheral blood and primary tumor tissues from LUSC patients before and after immunochemotherapy. Here, we used single-cell multi-omics profiling of paired pre- and post-treatment tumor and blood samples to demonstrate that elevated levels of three monocyte-derived indicators, including native circulating monocytes, native tumor classical monocytes, and APOBEC3A+

monocytes across both compartments, were strongly associated with pathologic responses to immunochemotherapy (IC) in patients with LUSC, validated by more independent patients' samples. These findings advanced prior work implicating monocytes as key modulators of tumor–immune interactions across cancers (Sheng et al, 2017; Schmall et al, 2015; Hanna et al, 2015) and extended current biomarker research beyond tumor-restricted measurements such as PD-L1 or TMB, which remain limited by spatial heterogeneity and sampling variability in clinical settings (Yu et al, 2019; Chan et al, 2019; Rakaee et al, 2023). By integrating paired pre- and post-treatment samples from peripheral blood and tumors, our findings supported a systemic–intratumoral axis through which monocyte composition reflects IC responsiveness in LUSC patients, offering a clinically accessible strategy for LUSC patient selection and monitoring.

The role of monocytes in shaping IC efficacy might be explained by their plasticity in differentiating into macrophages or dendritic cells, thus influencing antigen presentation, immune activation, and T-cell fitness within the LUSC microenvironment (Chen et al, 2023; Olingy et al, 2019; Elewaut et al, 2025). Notably, APOBEC3A+ monocytes were enriched in responders, suggesting a functional link with APOBEC-associated genomic instability, which has previously been associated with enhanced immunogenicity and improved responses to immunotherapy in multiple cancers (Petljak et al, 2022; Jalili et al, 2020; Gupta et al, 2025; Yang et al, 2024b). Our SCENIC-based regulatory network analysis further identified IRF7 as a putative upstream regulator of APOBEC3A in monocytes, consistent with IRF-mediated transcriptional programs in antiviral responses and type-I interferon signaling (Fig. EV5D–F) (Kubota et al, 2008; Romieu-Mourez et al, 2006). These data collectively implied that APOBEC3A+ monocytes might not simply serve as correlative indicators, but could participate in remodeling immune landscapes during IC treatment, warranting deeper functional investigation.

Interestingly, unlike observations in other cancer types, we did not find a consistent association between TCR expansion patterns and IC outcomes in LUSC, despite IC-induced CD8 + T-cell clonal expansion in some patients (Zhang et al, 2018; Liu et al, 2025). The heterogeneity of TCR responses observed across poor responders highlighted the complexity of immune activation in LUSC and reinforced the notion that T-cell expansion alone was insufficient to predict treatment success (Ma et al, 2025). These differences further underscore the need to evaluate immune dynamics beyond canonical adaptive markers, particularly in tumor entities

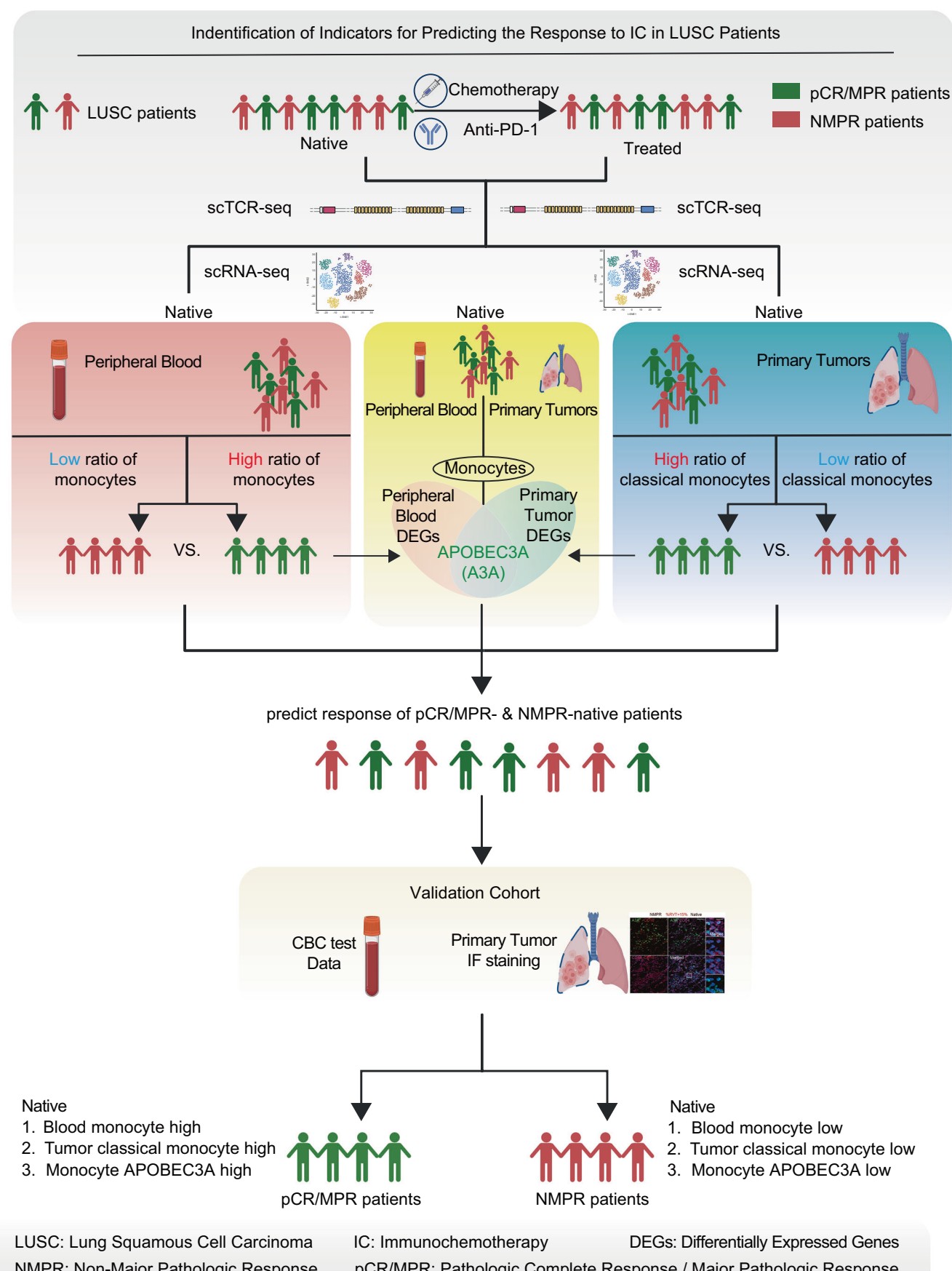

◄ **Figure 4. Working model.**

A comprehensive screening workflow was performed based on matched pre- and post-treatment peripheral blood and primary tumor samples from patients with LUSC. Three indicators were identified and validated to be associated with treatment responsiveness before treatment: the proportion of monocytes in peripheral blood, the proportion of classical monocytes in primary tumors, and the proportion of APOBEC3A+ monocytes in primary tumors. Patients with higher levels of these indicators exhibited significantly better therapeutic responses.

characterized by limited actionable mutations and lower immune infiltration like LUSC.

Together, our findings underscore the translational potential of monocyte-focused assessment as a minimally invasive approach to guide treatment stratification, capturing immunological states at the intersection of systemic and local tumor environments. This paired-sampling strategy, rarely applied in previous IC biomarker studies, may accelerate predictive indicator discovery and deepen mechanistic understanding of IC responsiveness in LUSC.

## Limitations of the study

Although APOBEC3A+ monocytes correlated with IC response, their exact functional roles remained undefined. Single-cell transcriptomic analyses alone could not confirm whether APO-BEC3A contributed directly to enhanced antitumor immunity or reflected an antiviral-like activation state driven by interferon responses (Vitiello et al, 2021). In addition, interactions between this monocyte subset and effector immune cells were not fully resolved by computational analyses, and future studies employing lineage tracing or spatial multi-omics will be necessary to clarify their functional relevance. Moreover, whether APOBEC3A+ monocytes represent a tumor-induced phenotype or a physiologic subset requires comparison with monocytes from healthy donors. Investigations using matched control cohorts will be essential to determine whether these cells constitute an adaptation to tumor-driven inflammatory cues or reflect broader immunological programming.

## Methods

### Patient cohorts and sample collection

scRNA-seq and single-cell T cell receptor sequencing (scTCR-seq) samples were obtained from 15 patients with LUSC at the Second Affiliated Hospitals of Zhejiang University School of Medicine and Fourth Affiliated Hospitals of Zhejiang University School of Medicine. Lung tumor sections for IF staining was obtained from LUSC patients at Second Affiliated Hospitals of Zhejiang University School of Medicine, Fourth Affiliated Hospitals of Zhejiang University School of Medicine, Taizhou Hospital of Zhejiang Province of Wenzhou Medical University, and Zhejiang Cancer Hospital. Ethical approval was granted by both institutions (Second Affiliated Hospital of Zhejiang University School of Medicine: reference 2024-1241; Fourth Affiliated Hospital of Zhejiang University School of Medicine: reference K2022179; Taizhou Hospital of Zhejiang Province of Wenzhou Medical University: reference 2024036; Zhejiang Cancer Hospital: reference IRB-2025-752(IIT)). Written informed consent was obtained from all patients. The patients received 2–3 cycles of neoadjuvant immune

checkpoint blockade combined with chemotherapy, followed by surgery. Peripheral blood samples were collected before and after treatment. Tumor samples from untreated patients were collected via diagnostic needle biopsy or bronchoscopy, whereas samples from treated patients were obtained surgically. Clinical details are provided in Table EV1. The investigators confirmed that the experiments conformed to the principles set out in the WMA Declaration of Helsinki and the Department of Health and Human Services Belmont Report.

### Ethics declarations

This study was approved by the Fourth Affiliated Hospital of Zhejiang University School of Medicine (reference number: K2022179), the Second Affiliated Hospital of Zhejiang University School of Medicine (reference number: 2024-1241), Taizhou Hospital of Zhejiang Province of Wenzhou Medical University (reference number: 2024036), and Zhejiang Cancer Hospital (reference number: IRB-2025-752(IIT)). All patients provided written informed consent.

### Graphics

Figure 1A, Fig. 4, Figure EV4E and part of Synopsis image were created with Biorender (https://www.biorender.com).

### IF (immunofluorescence) staining

IF staining of lung tumor sections was used to validate the differential APOBEC3A expression in monocytes across patient groups. Tissue sections were deparaffinized, rehydrated, and underwent antigen retrieval using 1× TE buffer for 10 min at 100 °C. The sections were blocked with IF buffer (1% BSA, 0.1% Triton X-100 in phosphate-buffered saline (PBS), and 1:2000 PROCLEAN) for 1 h at room temperature. Primary antibodies were incubated overnight at 4 °C, and they included anti-CD14 (17000-1-AP, 1:200; Proteintech), anti-CD16 (16559-1-AP, 1:200; Proteintech), and anti-APOBEC3A (25084-1-AP, 1:200; Proteintech). CD14, CD16, and APOBEC3A co-staining was performed using a four-color multiplex fluorescence kit (Hunan Aifang Biotechnology) through tyramide signal amplification technology, following the manufacturer's guidelines, and nuclei were counterstained with 4',6-diamidino-2-phenylindole (Solarbio C0060, 1:1000). Slides were mounted with Fluoromount-G® (SouthernBiotech) (Dakin et al, 2018; Mazambani et al, 2025; Nwabo Kamdje et al, 2023; Wu et al, 2023; Xu et al, 2026; Zerbino et al, 2018).

### Complete blood count test data collection

All patient information was routinely collected and documented by trained research coordinators throughout the entire study period.

The following data were obtained from medical records for all participants: age, sex, smoking history, comorbidities, laboratory results (including complete blood count, blood biochemistry, and tumor markers), chest radiography findings, and PD-L1 expression status. Three experienced pathologists (B.L., X.X., and S.J.) independently evaluated pathological responses according to a standardized protocol (Travis et al, 2020). Major pathological response (MPR) was defined as the presence of ≤10% viable tumor cells in the primary tumor site, assessed via residual viable tumor (RVT) in resected specimens; this included cases with no residual primary tumor but persistent nodal metastases. Complete pathological response (CPR) was defined as the absence of viable tumor cells in both the primary site and lymph nodes.

### Sample preparation for scRNA-seq and scTCR-seq

Fresh tumor tissues were stored in sCelLive™ Tissue Preservation Solution (Singleron) on ice after surgery. Tissues were washed with Hanks Balanced Salt Solution and enzymatically digested using sCelLive Dissociation Solution with the Singleron PythoN Tissue Dissociation System. Cell suspensions underwent red blood cell lysis using GEXSCOPE® RCLB (Singleron). Peripheral blood mononuclear cells were isolated through density gradient centrifugation and washed in Ca/Mg-free PBS, and red blood cells were lysed in a similar manner. Single-cell suspensions for scTCR-seq were prepared using Poisson distribution-based microfluidic devices.

### Library preparation for scRNA-seq and scTCR-seq

For scRNA-seq, cell suspensions were processed using the Singleron Matrix® Single Cell Processing System. The barcoded beads captured mRNA, generating cDNA libraries via reverse transcription and PCR amplification. The sequencing adapters were ligated after fragmentation. Libraries were prepared using GEX-SCOPE® Single Cell RNA Library Kits and sequenced on Illumina NovaSeq 6000 (150 bp paired-end reads). For scTCR-seq, libraries were constructed using the sCircle® Single Cell Full-Length Immuno_TCR/BCR Library Kit (Singleron), involving poly(A) tail capture, reverse transcription, enrichment of full-length TCR cDNA, and fragmentation. The libraries were sequenced in a similar manner.

### Preprocessing and quality control of sequencing data

The scRNA-seq and TCR-seq data were processed using CeleScope v1.9.0 pipeline (https://github.com/singleron-RD/CeleScope). Low-quality reads and adapter sequences were removed using Cutadapt v1.17, and alignment to the reference genome GRCh38 (Ensembl v92 annotation) was performed using STAR v2.6.1a (Martin, 2011; Zerbino et al; Dobin et al, 2013). Subsequent analyses involved using Seurat (R) and Scanpy (Python) (Hao et al, 2024; Wolf et al, 2018; Hao et al, 2021). Cells with <200 genes or genes expressed in <3 cells were excluded. Potential doublets were removed using the DoubletFinder v2.0.4 (McGinnis et al, 2019). The doublet rate was set 7.5%. According to the pipeline and doublet rate estimation table provided by the author of DoubletFinder, 10,000 recovered cells corresponding to 7.5–7.6% doublet rate (Table EV3). Because the recovered cell numbers of most scRNA-seq samples fluctuated nearby 10,000 cells, we set 7.5% as the doublet rate.

### Integration and dimensionality reduction of scRNA-seq data

Data from different samples were integrated using atomic sketch integration and SCTransform (Seurat v4.1.9001). Batch effects were minimized by excluding hemoglobin, ribosomal and mitochondrial genes, and *MALAT1* from the integration steps. Dimensionality reduction was performed using UMAP visualization based on integrated embeddings (Seurat v4.1.9001) (Hao et al, 2021).

Preprocessed scRNA-seq of published LUSC tumors and lung without tumor were downloaded from https://zenodo.org/records/7227571 (Salcher et al, 2022).

### Clustering and annotation

The cells underwent unsupervised clustering using the Leiden algorithm (Seurat v5.1.0), informed by SingleR annotations and marker gene expression. Tumor cells were identified using LUSC-specific markers (*KRT5*, *KRT6A*, *TP63*, and *DSG3*) and validated through comparison with previously published datasets (Salcher et al, 2022). The annotation of tumor cells was validated by copy number variation (CNV) analysis (detailed in the CNV analysis section). Monocyte subsets, including APOBEC3A+/− populations, were further refined using scANVI embedding (Scanpy v1.10.0; Python v3.10.6). Neutrophils were excluded because of their inconsistent representation (Fig. EV1F). Major cell-type annotations were validated through comparison with published single-cell RNA-seq datasets from LUSC tumors (Fig. EV1G).

### CNV analysis

CNV scores of tumor cells were calculated using InferCNV v1.14.1 (https://github.com/broadinstitute/infercnv). Tumor cells from primary tumor samples were confirmed by identifying large-scale genomic CNVs using non-tumor cells as controls.

### Cell communication analysis

CellChat v1.6.1 was used to assess cell–cell interactions, and the results were visualized using heatmaps and bubble plots (Jin et al, 2021, 2023). The "Relative values" in Figs. EV2A, EV3C and EV5A represent the cell–cell interaction strength value between monocytes and other cells in pCR/MPR patients relative to NMPR patients. Positive "Relative values" means the cell–cell interaction strength value between monocytes and other cells in pCR/MPR patients is higher than that in NMPR patients, and vice versa.

### Differential gene expression and pathway analysis

Differentially expressed genes and pathway enrichments were identified and visualized using SCP v0.5.6. GO enrichment analysis was conducted with parameter "DE_threshold = 'avg_log2FC > log2(0.5) & p_val_adj < 0.05', db = 'GO_BP'". GSEA enrichment analysis was conducted with parameter "DE_threshold = 'p_val_adj < 0.05', scoreType = 'std', db = c('GO_BP')".

## Imitating case-deletion diagnostics

To ensure the precision of identified indicators, we remove outlier and influential samples by imitating case-deletion diagnostics (Christensen et al, 1992; De Bastiani et al, 2018). We generated DEGs of pCR/MPR versus NMPR monocytes in native blood and tumors with one sample removed each time. Then we did PCA dimensionality reduction of the DEG fold change profiling of each deletion. If one profiling with the specific sample was outlier in the PCA, then we compared the pCR/MPR versus NMPR monocyte DEGs with no samples removed (Control) and with the specific sample removed (Test). If the top 100 upregulated and top 100 downregulated DEGs of Control and Test were similar, then the specific sample was reserved. If not, the sample was removed.

## Cell differentiation analysis

CytoTRACE2 (v1.1.0) was used to assess the differentiation states and potency levels of the monocyte subsets with varying APOBEC3A expression levels (Kang et al, 2025).

## TCR diversity and clonality analysis

The processed TCR matrices were integrated with the scRNA-seq data using scRepertoire v2.0.0 (Borcherding et al, 2020). Clonal diversity metrics (e.g., the Gini–Simpson index) were calculated using scRepertoire. TCR clonotypes were determined based on clonotype expansion levels: Single (one occurrence); Small (>1 and ≤5); Medium (> 5 and ≤20); Large (> 20 and ≤100); Hyperexpanded (>100).

## Calculation and validation of combination strategies

To identify the feasible combinations of indicators and their thresholds, we iterated thresholds of three indicators separately in scRNA-seq data of native peripheral blood and primary tumors with different combinations of three indicators. Because the value of three indicators were positively related to IC response, higher thresholds meant higher accuracy and lower coverage. To balance the accuracy and coverage, we used the ratio of accuracy versus the square root of the number of patients meeting the thresholds to evaluate the quality of combinations and thresholds. The accuracy and coverage of thresholds applied in scRNA-seq data were collected in Table EV7.

We validated the calculation result by applying the combinations and thresholds on CBC test data and IF staining results (Tables EV8 and EV9). The detailed ratio data of validation used in Table EV9 were listed in Table EV8.

## Quantification of IF staining results

### Screening logic
Let each patient $i = 1, \ldots, N$ have three quantitative indicators $x_{iA}, x_{iB}, x_{iC}$ corresponding to markers A, B, C, and a binary outcome $y_i \in \{0, 1\}$, where $y_i = 1$ denotes pCR/MPR and $y_i = 0$ denotes NMPR. For each indicator $k \in \{A, B, C\}$, we define a candidate threshold range $[L_k, U_k]$, where $L_k$ is the lower bound of $x_{ik}$ in the pCR/MPR group and $U_k$ is the upper bound of $x_{ik}$ in the NMPR group. Within these ranges, we systematically explore thresholds $\tau_k \in [L_k, U_k]$ and all allowed combinations of indicators

$S \in \{\{A\}, \{B\}, \{C\}, \{A, B\}, \{A, C\}, \{B, C\}\}$. For a given combination $S$ and threshold vector $\boldsymbol{\tau}_S = (\tau_k)_{k \in S}$, a patient is predicted as pCR/MPR if and only if $x_{ik} \geq \tau_k$ for all $k \in S$; otherwise, the patient is classified as NMPR or left unassigned, depending on the chosen implementation. For each $(S, \boldsymbol{\tau}_S)$, we compute prediction coverage (the proportion of patients receiving a prediction) and prediction accuracy (the proportion of correct predictions among those covered), and select the combination of indicators and thresholds that best balances these two quantities according to a pre-specified criterion. The specific choice reported in this study represents one concrete instance of this general framework.

### Multi-channel logical gating of positive cell
For each multi-channel immunofluorescence image, nuclei were first segmented on the DAPI channel and each nucleus $j$ in image $i$ was associated with three marker channels: CD14 (ch1), CD16 (ch2), and APOBEC3A (ch3). For every channel $c \in \{1, 2, 3\}$, we constructed a global positivity mask by applying rollingball background correction, estimating a channel-specific global intensity threshold $T_c$ from the corrected intensity distribution (here using the 50th percentile), and binarizing the image followed by removal of small objects and optional morphological closing. Given the nuclear mask of cell $j$, we then defined two binary predicates per channel: (i) a perinuclear "outerring" predicate $O_c(i, j) \in \{0, 1\}$, obtained by dilating and eroding the nucleus to form an annular ring, sampling this ring into angular bins, and declaring $O_c(i, j) = 1$ if the fraction of bins containing sufficient positive pixels exceeded a preset coverage threshold and the largest contiguous negative angular gap was below a preset maximum; and (ii) a nuclear-overlap predicate $N_c(i, j) \in \{0, 1\}$, defined as $N_c(i, j) = 1$ if the fraction of nuclear pixels positive for channel $c$ was greater than a fixed minimal fraction (e.g., 0.5). On top of these predicates, we defined a family of Boolean aggregation rules $A_k$ to gate positive cell as

$$A1(i, j) = O_2(i, j) \wedge \neg(O_1(i, j) \vee N_1(i, j))$$
$$A2(i, j) = O_2(i, j) \vee O_1(i, j) \vee N_1(i, j)$$
$$A3(i, j) = (N_3(i, j) \wedge O_3(i, j)) \wedge (O_2(i, j) \vee O_1(i, j) \vee N_1(i, j))$$
$$A4(i, j) = N_3(i, j) \wedge O_3(i, j)$$

where $\wedge$, $\vee$, and $\neg$ denote logical AND, OR, and NOT, respectively. This logical-gating formalism provides a general multi-channel framework; the four rules $A1$, $A2$, $A3$, and $A4$ reported in this study are specific instantiations within this rule family.

## A priori sample-size assessment for the independent validation cohort

In neoadjuvant lung cancer studies, obtaining high-quality tumor tissue (sufficient material, pre-analytic handling, and assay success) is typically more constrained than obtaining peripheral blood. Accordingly, we prioritized tumor-derived indicator validation as the primary inferential objective and planned the cohort size to achieve adequate statistical power within a logistic-regression framework.

### Planned analysis model
The primary endpoint was binary pCR/MPR status ($Y = 1$ for pCR/MPR, $Y = 0$ otherwise). Each candidate tumor indicator was modeled as a continuous predictor and standardized using z-score

transformation (mean 0, standard deviation 1), such that the odds ratio (*OR*) reflects the change in odds of pCR/MPR per 1-standard-deviation ($1 - SD$) increase in indicator value. The planned single-covariate logistic regression model was:

$$logit\{Pr(Y = 1, |, X)\} = \alpha_0 + \sigma \cdot X, \text{ where } \sigma = ln(OR).$$

### A priori sample-size calculation

Sample-size planning was performed using the information-based approximation for logistic regression originally developed by Whittemore and summarized in the corrected and tabulated formulation by Hsieh (1989) for a single continuous covariate (Hsieh, 1989). Under this approach, the required total sample size (*N*) depends on (i) the targeted effect size $\sigma = \ln(OR)$, (ii) the anticipated event probability *P* at the mean covariate value (i.e., $X = 0$ after standardization), (iii) the two-sided type I error rate *α*, and (iv) power ($1 - \beta$). The continuous-covariate correction term described by Hsieh was applied. Design inputs were specified as follows:

- Target effect size: Pilot analyses suggested a plausible range of $OR \in [2.0, 2.5]$; we therefore prespecified $OR_{target} = 2.3$ as a representative target effect size for study planning.
- Type I error and power: two-sided $\alpha = 0.05$ and power $(1 - \beta) = 0.80$.
- Event probability: anticipated pCR/MPR rate $P = 0.65$, consistent with published neoadjuvant immunotherapy/chemoimmunotherapy cohorts $P \in [0.6, 0.7]$ and aligned with our regimen and pilot observations (Jiang et al, 2022).

With these prespecified assumptions, the Hsieh approximation yielded a minimum required validation cohort size of approximately N ≈ 50 patients.

### Achieved independent validation cohort and inferential analyses

Guided by the above a priori rationale and within feasibility constraints, we ultimately assembled an independent tumor validation cohort of 54 patients, with an observed pCR/MPR proportion of 62.9%, meeting the prespecified sample-size requirement. Within this independent cohort, we evaluated two candidate tumor indicators, both analyzed as z-score–standardized continuous variables. Each indicator was assessed using univariable logistic regression, consistent with the single-covariate framework underlying the sample-size calculation. The proportion of classical monocytes among primary tumor monocytes showed an *OR* of 1.87 (95% CI 1.04–3.39; $p = 0.0379$) per $1 - SD$ increase, while the ratio of APOBEC3A+ monocytes among total monocytes in native primary tumors showed an *OR* of 2.29 (95% CI 1.07–4.94; $p = 0.0336$). These effect sizes were statistically significant and consistent with the prespecified design-stage target effect size ($OR_{target} = 2.3$).

In summary, the independent cohort used for tumor indicator validation was supported by an explicit a priori sample-size rationale, and the observed effect sizes at this sample scale were consistent with the prespecified design assumptions, supporting the robustness of the primary validation findings. Moreover, the validation cohort size we calculated is similar to those reported in other studies on response predictors of NSCLC treatment, which strengthened the reliability of our size estimation (Yeong et al, 2021

**The paper explained**

**Problem**

Immunotherapy combined with chemotherapy is a standard treatment for lung squamous cell carcinoma, but only some patients benefit. Clinicians currently lack simple tools to predict who will respond before treatment begins.

**Results**

Using single-cell analysis of blood and tumor samples collected before and after treatment, we found that patients who responded well had higher levels of monocytes in blood and tumors. A specific monocyte population marked by APOBEC3A was consistently enriched in responders. These findings were confirmed using routine blood tests and standard tissue staining.

**Impact**

Our results suggest that simple blood and tissue markers could help identify patients most likely to benefit from immunochemotherapy, supporting more personalized treatment decisions and reducing unnecessary toxicity.

($n = 35$); Kumagai et al, 2020 ($n = 48$); Gettinger et al, 2018 ($n = 49$)).

## Quantification and statistical analysis

The statistical comparison of cell-type percent or subpopulation percent was conducted using two-sided Wilcoxon test. The statistical comparison of APOBEC3A expression between tumor sections from different response groups was conducted using two-sided t-test. Asterisks indicate the degree of significance by *p* values: NS: no significant difference, $*p < 0.05$, $**p < 0.01$, $***p < 0.001$.

## Data availability

The raw data of scRNA-seq and TCR-seq can be accessed at the Genome Sequence Archive at the National Genomics Data Center (Beijing, China) under the ID HRA010788 and HRA009058.

The source data of this paper are collected in the following database record: biostudies:S-SCDT-10_1038-S44321-026-00410-y.

## Peer review information

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

## Acknowledgements

This work was partially supported by grants to Dr. Kai Wang from the National Key Research and Development Program of China (2024YFA1108500), National Natural Science Foundation of China (U23A20467). This work was partially supported by grants to Dr. Jian Liu from the Natural Science Foundation (NSF) of China (General Grant: 82172899 and 82472637), the NSF of Zhejiang Province (Continuation Grant of Distinguished Young Scholars: RG26H160003; Distinguished Young Scholars: LR22H160002), and the Noncommunicable Chronic Diseases-National Science and Technology Major Project (2023ZD0502900/2023ZD0502902 and 2023ZD0507500/2023ZD0507501). This work was partially supported by grants to Dr. Hai Song from Key R&D Program of Zhejiang (Grant No. 2025C02091), Medical Scientific Research Foundation of Zhejiang Province (Grant No. WKJ-ZJ-2508). This work was partially supported by grants to Dr. Pingli Wang from Noncommunicable Chronic Diseases-National Science and Technology Major Project (2024ZD0529300). We thanked Yunze Wang's help on the calculation.

## Author contributions

**Jiangnan Zhao**: Conceptualization; Resources; Writing—review and editing. **Zijin Wang**: Conceptualization; Formal analysis; Investigation; Methodology; Writing—original draft; Writing—review and editing; Bioinformatics. **Manyu Xiao**: Investigation; Methodology; experiments. **Yeyuan Zhang**: Investigation; Clinical sample collection, and analyses. **Boxin Liu**: Investigation; Methodology; Bioinformatics. **Silin Chen**: Investigation; Visualization; experiments. **Yiping Tian**: Investigation; Experiments. **Dongqing Lv**: Investigation; Experiments. **Pingli Wang**: Investigation; Experiments. **Hai Song**: Writing—review and editing. **Yuefeng Wu**: Writing—review and editing. **Jian Liu**: Conceptualization; Resources; Supervision; Funding acquisition; Investigation; Writing—original draft; Writing—review and editing. **Kai Wang**: Conceptualization; Resources; Supervision; Funding acquisition; Investigation; Writing—review and editing.

Source data underlying figure panels in this paper may have individual authorship assigned. Where available, figure panel/source data authorship is listed in the following database record: biostudies:S-SCDT-10_1038-S44321-026-00410-y.

## Disclosure and competing interests statement

K Wang, J Liu, J Zhao, Z Wang, and M Xiao have a patent application for APOBEC3A diagnosis in the positive prediction of IC response.

# Expanded View Figures

**Figure EV1.   Cell-type annotation validation.**

(A) Box plot and comparisons of smoking index between single-cell sequenced pCR/MPR and NMPR patients. Box plots followed the Tukey style (patient $n = 15$, pCR/MPR $n = 8$, NMPR $n = 7$; Centre line: median. Box bounds: 25th and 75th percentiles. Whiskers: extending to the most extreme data points within 1.5 times the interquartile range (IQR) from the box bounds. Outliers: points beyond the end of the whiskers). (B) UMAP of scRNA-seq data before removing batch effect. (C) Dot plot displaying marker gene expression used for major cell-type annotation. (D) Copy number variation scores compared between suspected tumor cells and other cells in the peripheral blood (blood sample $n_{blood} = 23$, before treatment $n_{pCR/MPR} = 5$, $n_{NMPR} = 4$, after treatment $n_{pCR/MPR} = 7$, $n_{NMPR} = 7$) and primary tumors (tumor sample $n_{tumor} = 26$, before treatment $n_{pCR/MPR} = 7$, $n_{NMPR} = 5$, after treatment $n_{pCR/MPR} = 7$, $n_{NMPR} = 7$). Comparison was conducted using Wilcoxon test (significant $p < 0.05$). Box plots followed the Tukey style (Centre line: median. Box bounds: 25th and 75th percentiles. Whiskers: extending to the most extreme data points within 1.5 times the interquartile range (IQR) from the box bounds. Outliers: points beyond the end of the whiskers). (E) IF staining of KRT5 (green) and DAPI (blue) of native tumor slices from patients in scRNA-seq cohort. (F) Box plot of major cell type proportion across different response groups (blood sample $n_{blood} = 23$, before treatment $n_{pCR/MPR} = 5$, $n_{NMPR} = 4$, after treatment $n_{pCR/MPR} = 7$, $n_{NMPR} = 7$; tumor sample $n_{tumor} = 26$, before treatment $n_{pCR/MPR} = 7$, $n_{NMPR} = 5$, after treatment $n_{pCR/MPR} = 7$, $n_{NMPR} = 7$). Box plots followed the Tukey style (Centre line: median. Box bounds: 25th and 75th percentiles. Whiskers: extending to the most extreme data points within 1.5 times the interquartile range (IQR) from the box bounds. Outliers: points beyond the end of the whiskers). (G) Comparison of major cell-type proportions excluding tumor cells (sample $n = 49$) with published LUSC scRNA-seq data (sample $n = 113$) before treatment. Comparison was conducted using Wilcoxon test (significant $p < 0.05$). Box plots followed the Tukey style (Centre line: median. Box bounds: 25th and 75th percentiles. Whiskers: extending to the most extreme data points within 1.5 times the interquartile range (IQR) from the box bounds. Outliers: points beyond the end of the whiskers).

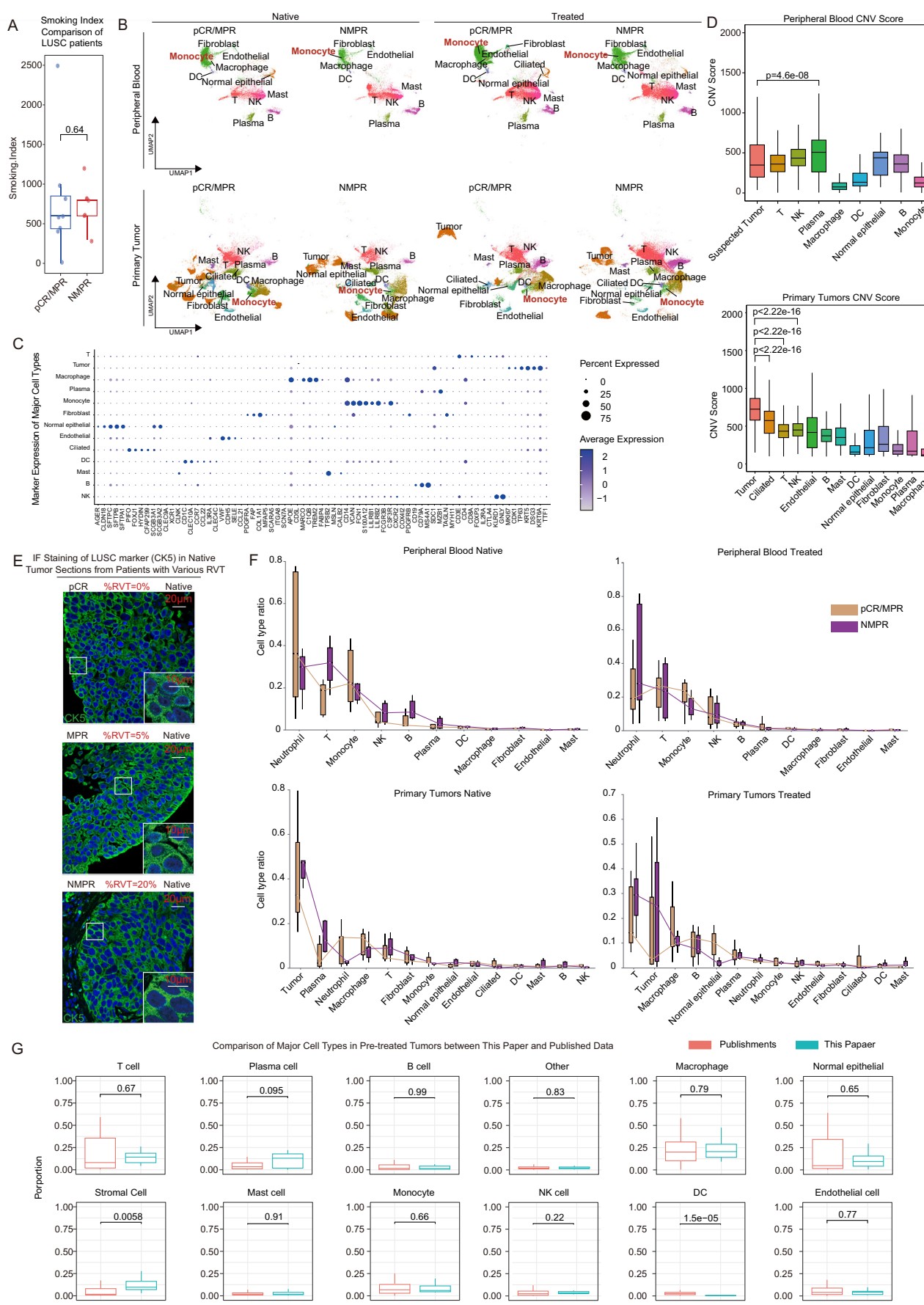

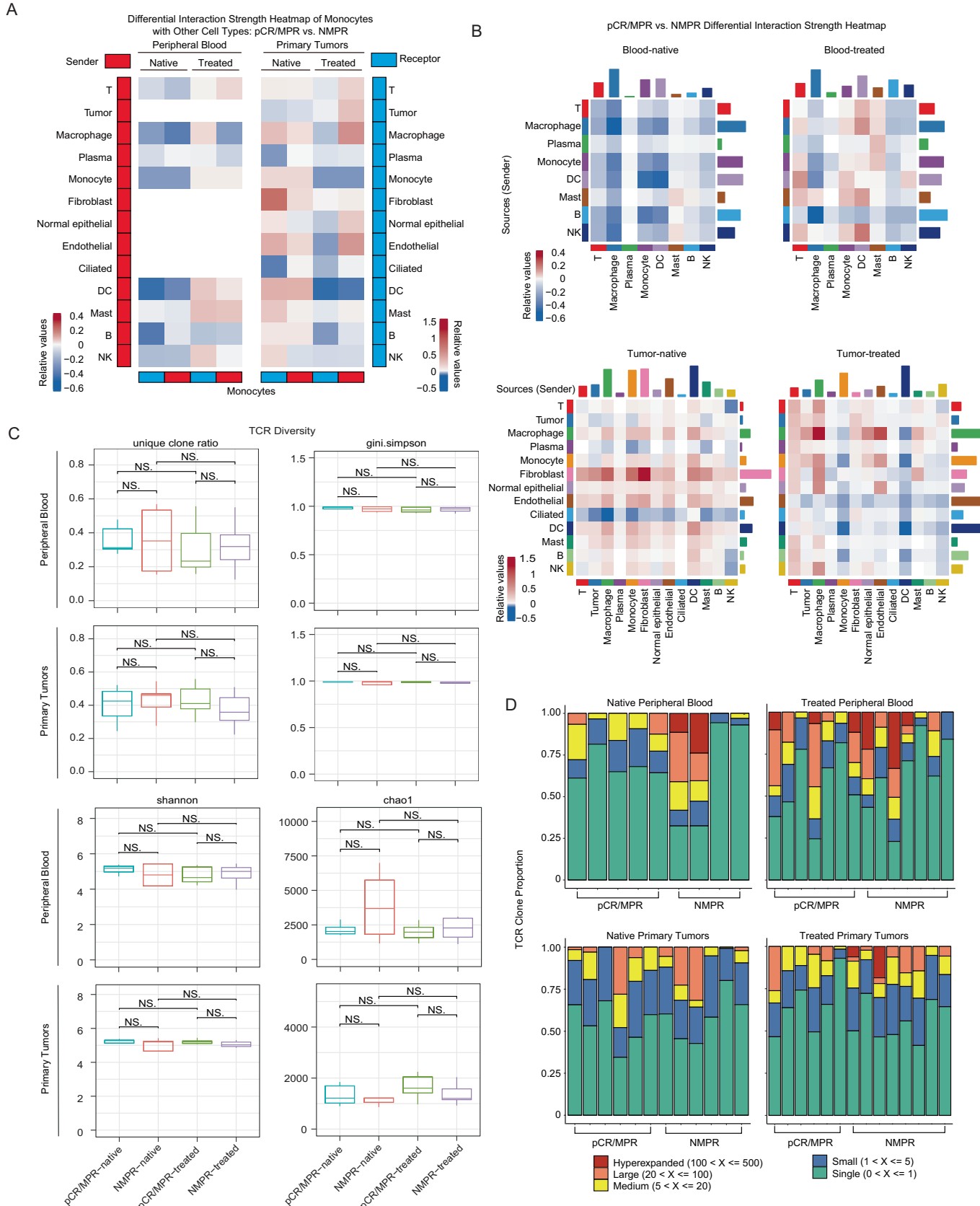

**Figure EV2. Cell–cell communication heatmap and TCR diversity.**

(A) pCR/MPR vs. NMPR differential interaction strength heatmap of major cell types. (B) Cell–cell communication difference heatmap among monocytes and other major cell types with pCR/MPR versus NMPR. (C) Comparison of four TCR diversity indicators among different response groups and between pre- and post-treatment groups (blood sample $n_{blood} = 23$, before treatment $n_{pCR/MPR} = 5$, $n_{NMPR} = 4$, after treatment $n_{pCR/MPR} = 7$, $n_{NMPR} = 7$; tumor sample $n_{tumor} = 26$, before treatment $n_{pCR/MPR} = 7$, $n_{NMPR} = 5$, after treatment $n_{pCR/MPR} = 7$, $n_{NMPR} = 7$). Comparison was conducted using Wilcoxon test (significant $p < 0.05$). Box plots followed the Tukey style (Centre line: median. Box bounds: 25th and 75th percentiles. Whiskers: extending to the most extreme data points within 1.5 times the interquartile range (IQR) from the box bounds. Outliers: points beyond the end of the whiskers). (D) Relative proportions of specific TCR clonotypes across samples; TCR clonotypes were determined based on clonotype expansion levels: Single (one occurrence); Small (>1 and ≤5); Medium (>5 and ≤20); Large (>20 and ≤100); Hyperexpanded (>100).

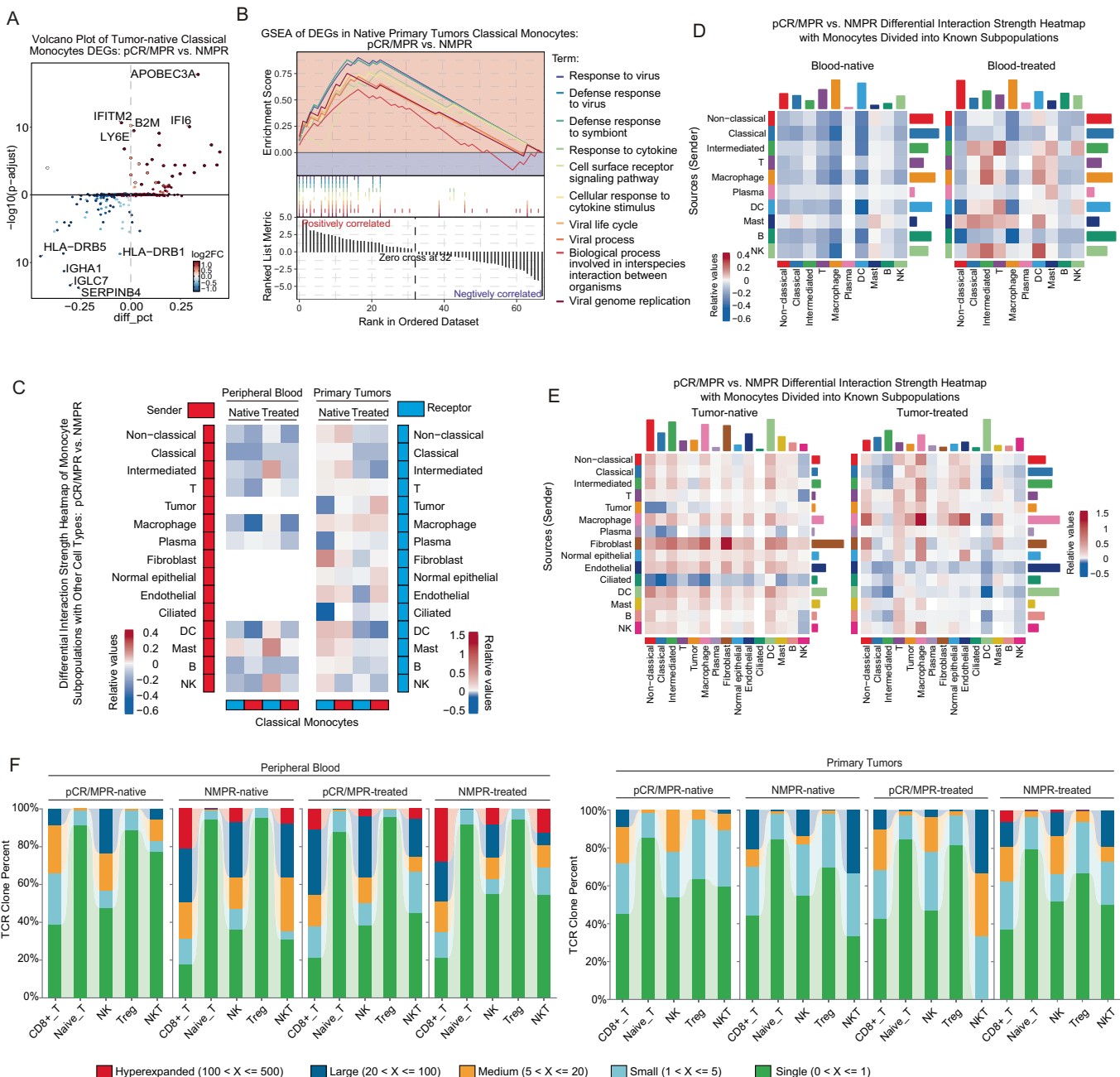

**Figure EV3. GSEA and cell–cell communication of monocyte subpopulations before and after treatment.**

(A) Volcano plot of DEGs with pCR/MPR versus NMPR classical monocytes in native tumors (native tumor sample $n_{tumor} = 12$, $n_{pCR/MPR} = 7$, $n_{NMPR} = 5$; the x-axis indicates percentage differences in gene-expressing cells between groups, and the y-axis shows $-\log_{10}$(adjusted p-value). Dots are colored based on $\log_2FC$ values. Two-sided Wilcoxon test was utilized to calculate p-values and p-values were adjusted by Bonferroni correction. (B) GSEA plot of top 10 upregulated pathways based on DEGs in Fig. EV3A. (C) Cell–cell communication difference heatmap among classical monocytes and other major cell types with pCR/MPR versus NMPR. (D) Peripheral blood pCR/MPR vs. NMPR differential interaction strength heatmap of major cell types with monocytes divided into three subpopulations. (E) Primary tumor pCR/MPR vs. NMPR differential interaction strength heatmap of major cell types with monocytes divided into three subpopulations. (F) Relative proportions of specific TCR clonotypes among T cell subpopulations across patient groups; TCR clonotypes were determined based on clonotype expansion levels: Single (one occurrence); Small (>1 and ≤5); Medium (>5 and ≤20); Large (>20 and ≤100); Hyperexpanded (>100); T cell subpopulation abbreviations: CD8 + T$_{EMRA}$/T$_{EFF}$: effector memory or effector T cells; CD8 + T$_{CM}$: CD8+ central memory T cells; T$_N$: naive T cells; CD4+ Normal-T$_{CM}$: CD4$^+$ Normal-central memory T cells; CD4+ Blood-T$_{CM}$: CD4$^+$ Blood-central memory T cells; Tumor-T$_{reg}$: tumor-infiltrating T regulatory; CD8 + T$_{EX}$: exhausted CD8$^+$ T cells; CD4 + T$_{RM}$: tissue-resident memory T cells.

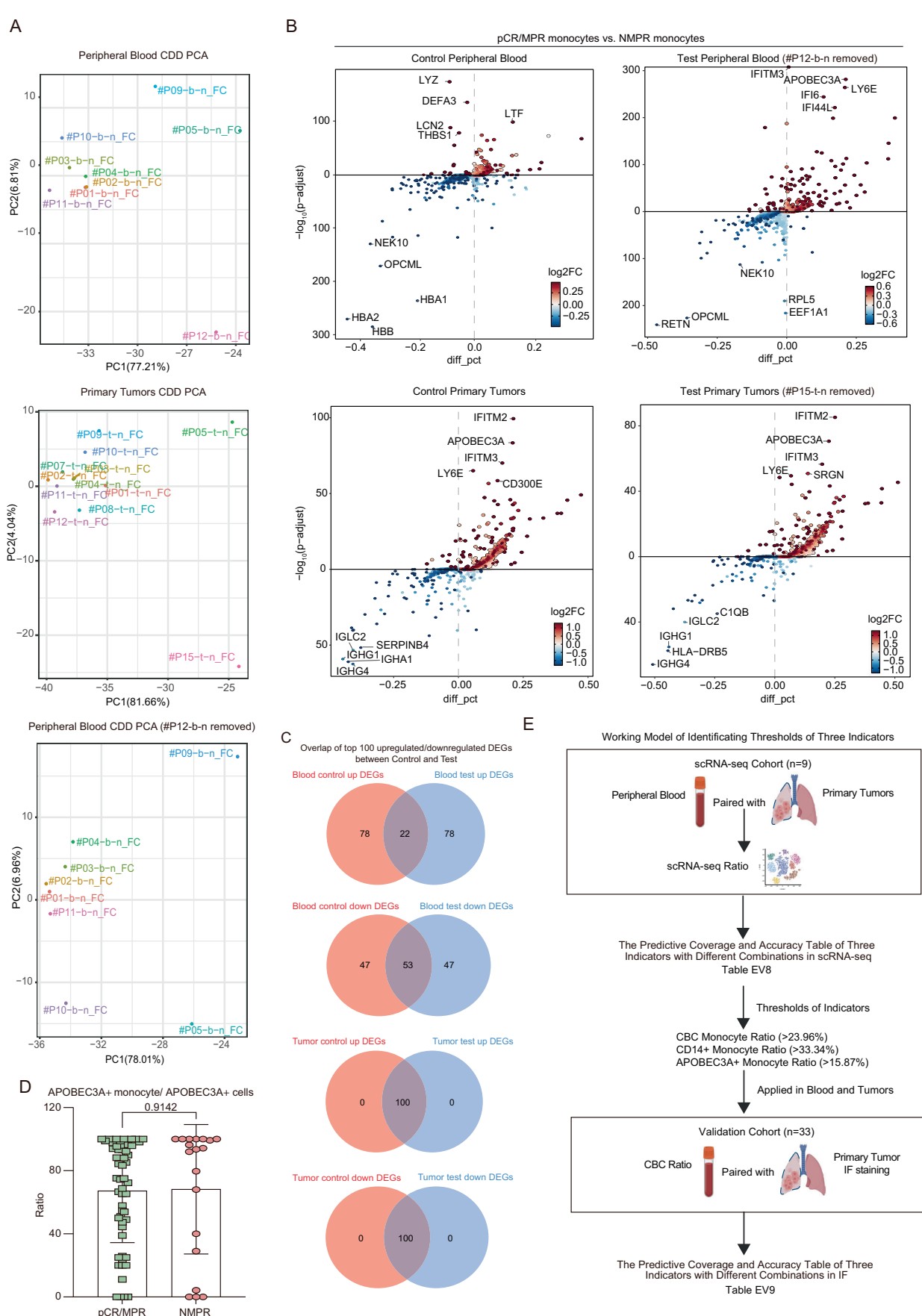

◀ **Figure EV4.   Imitating case deletion diagnosis analysis of native peripheral blood and primary tumors.**

(A) PCA dimensionality reduction of imitating case deletion diagnosis (CDD) in all native blood, all native tumor and native blood samples with #P12-b-n sample removed. Each point indicates the DEG fold change profiling of native pCR/MPR versus NMPR monocytes after one sample removed. The points are named by the removed samples. Fox example, P12-b-n_FC means that we removed #P12-b-n sample from pCR/MPR native blood and calculated DEGs fold changes with rest pCR/MPR versus NMPR native blood monocytes. (B) Volcano plot of DEGs with native pCR/MPR versus NMPR monocytes. Control means all native pCR/MPR versus NMPR samples (native blood sample $n_{blood} = 9$, $n_{pCR/MPR} = 5$, $n_{NMPR} = 4$; native tumor sample $n_{tumor} = 12$, before treatment $n_{pCR/MPR} = 7$, $n_{NMPR} = 5$). Test means all pCR/MPR versus NMPR with outlier samples removed (removed sample ID in title; native blood sample $n_{blood} = 8$, $n_{pCR/MPR} = 5$, $n_{NMPR} = 3$; native tumor sample $n_{tumor} = 11$, before treatment $n_{pCR/MPR} = 7$, $n_{NMPR} = 4$). Two-sided Wilcoxon test was utilized to calculate *p*-values and *p*-values were adjusted by Bonferroni correction. (C) Top 100 upregulated and top 100 downregulated DEGs overlap between Control and Test results in (B). (D) The ratio of APOBEC3A+ monocyte versus APOBEC3A+ cells based on IF staining (pCR/MPR $n = 34$, NMPR $n = 20$; two-sided T-test, significant $p < 0.05$). Error bars of bar plots represented the standard deviation. (E) Working model of identifying thresholds of three indicators.

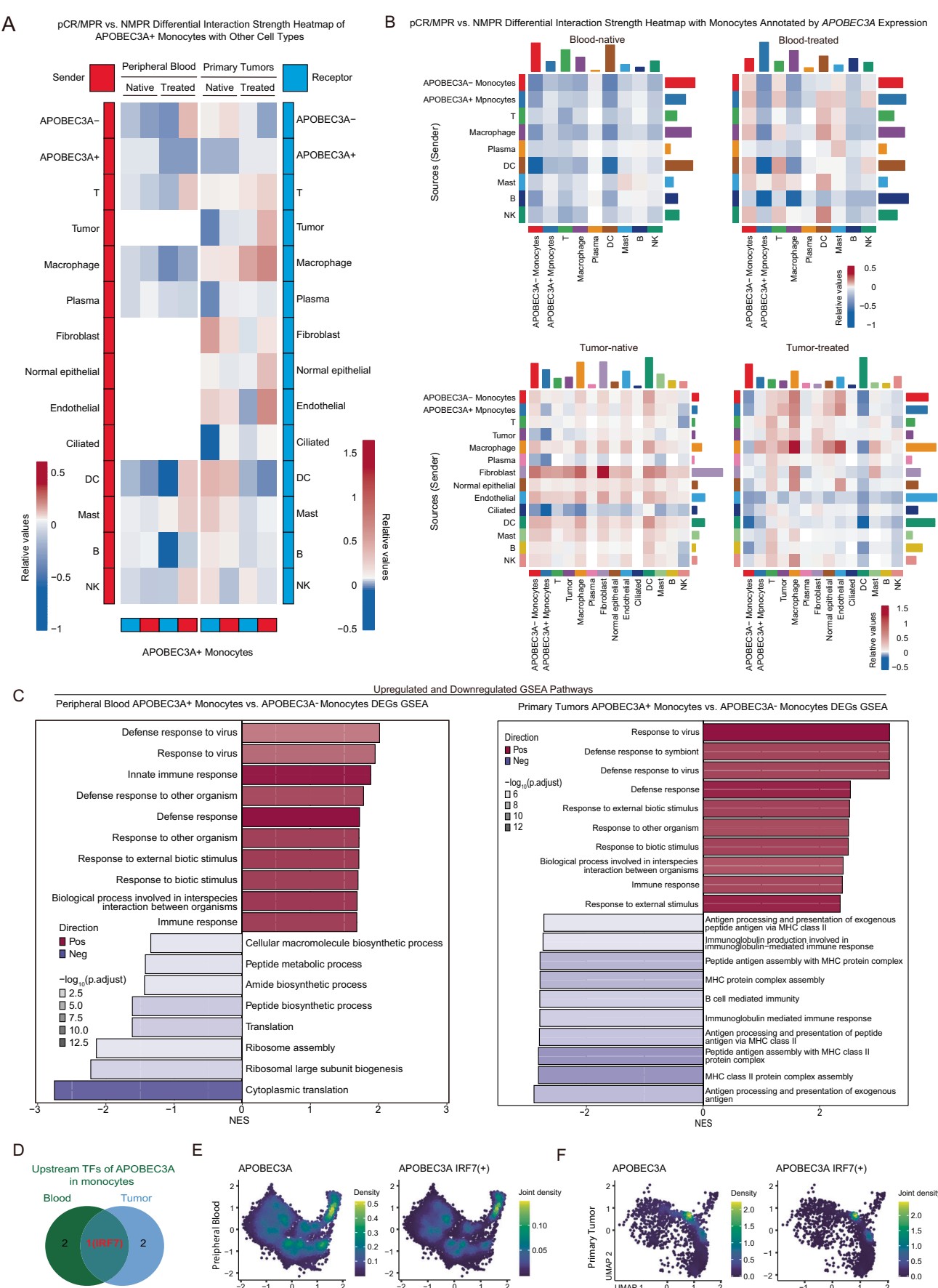

**Figure EV5. GSEA analysis, cell–cell interaction and upstream analysis of APOBEC3A+/- monocytes of LUSC patients.**

(A) Cell–cell communication difference heatmap among APOBEC3A+/- monocytes and other major cell types with pCR/MPR versus NMPR. (B) pCR/MPR vs. NMPR differential interaction strength heatmap with monocytes annotated by APOBEC3A expression. (C) GSEA pathway bar plot showing upregulated and downregulated DEGs of APOBEC3A+ versus APOBEC3A- monocytes in blood and tumors. (D) Overlap of upstream TFs of APOBEC3A in native blood and tumor monocytes. (E) Expression density of APOBEC3A; Co-expression density of APOBEC3A and IRF7(+) in native blood monocytes. (F) Expression density of APOBEC3A; Co-expression density of APOBEC3A and IRF7(+) in native tumor monocytes.

