## [Peer Review File · EMBO Molecular Medicine]

A Monocyte-centered Framework for Predicting Immunochemotherapy Efficacy in LUSC Patients

Jiangnan Zhao, Zijin Wang, Manyu Xiao, Yeyuan Zhang, Boxin Liu, Silin Chen, Yiping Tian, Dongqing Lv, Pingli Wang, Hai Song, Yuefeng Wu, Jian Liu, and Kai Wang

Corresponding authors: Kai Wang (Kaiw@zju.edu.cn) , Jian Liu (JianL@intl.zju.edu.cn)

Review Timeline:

Submission Date:	5th Jun 25
Editorial Decision:	25th Jun 25
Appeal:	12th Dec 25
Editorial Decision:	13th Jan 26
Revision Received:	24th Jan 26
Editorial Decision:	17th Feb 26
Revision Received:	20th Feb 26
Accepted:	9th Mar 26

Editor: Lise Roth

Transaction Report:

24th Jun 2025

Decision on your manuscript EMM-2025-22055

Dear Prof. Wang,

Thank you for submitting your manuscript to EMBO Molecular Medicine. We have now received the reports from the two reviewers who have evaluated your manuscript. As you will see from the reports below, they recognize the interest of the findings, but they also raise significant concerns, including -but not limited to- small sample size, lack of functional validation, lack of mechanistic insight, etc.

Due to the nature of the reviewers' concerns and the amount of work that would be required to satisfactorily address them, and given that at EMBO Press we encourage a single round of revisions in a limited time frame, I am afraid that I have little choice but to return the manuscript to you at this time with the decision that we cannot offer to publish it.

While we cannot pursue this manuscript further, we encourage you to transfer your study to our not-for-profit open-access sister journal, Life Science Alliance (LSA). We shared your manuscript and the accompanying reviews with LSA Executive Editor, Tim Fessenden, who is interested in these findings. He is pleased to offer publication of this manuscript at LSA pending the following revisions:

- Resolve all points raised by Reviewer 1, in particular point 4, except new experimental data is not required to resolve point 2.
- Acknowledge limitations of this work and consider the additional points of discussion suggested by Reviewer 2. This includes softening claims of a novel biomarker and amending the title and abstract accordingly.

We encourage you to use the link below to transfer your manuscript to LSA. You do not need to revise the manuscript before transferring it to LSA. Once you transfer, Dr. Fessenden will email you an invitation to revise and resubmit, listing the same revision requests as mentioned above. Please feel free to reach out at t.fessenden@life-science-alliance.org if you have any questions about the LSA journal, the transfer process, or the revisions requested.

I am very sorry to disappoint you in this occasion, and hope you will view the opportunity to transfer to LSA favorably.

Sincerely,

Lise Roth

Lise Roth
Senior Editor
EMBO Molecular Medicine

Referee #1 (Remarks for Author):

The manuscript presents compelling novel findings by identifying APOBEC3A high monocytes in both native peripheral blood and primary tumors as a robust biomarker predicting better immunochemotherapy (IC) response in lung squamous cell carcinoma (LUSC) patients. This directly addresses a critical unmet clinical need for reliable predictive biomarkers in LUSC, where current approaches (e.g., PD-L1, TMB) have limited accuracy due to tumor heterogeneity. The use of multi-omics (scRNA-seq and scTCR-seq) on paired pre- and post-treatment samples from both peripheral blood and primary tumors in LUSC patients is methodologically rigorous and provides significant depth of insight into immune cell dynamics and biomarker discovery.

Overall, I consider this work to be a valuable contribution to the field. However, to enhance its impact and ensure the robustness of its conclusions, several key areas require clarification and further validation.

Major Comments

1. The study's findings are based on a cohort of 15 LUSC patients. While the single-cell depth is high, this is a relatively small number for biomarker validation, especially given the observed heterogeneity. Additional validation in larger, independent cohorts-either through scRNA-seq or targeted bulk approaches-would significantly strengthen the generalizability and clinical relevance of the findings.

2. The study successfully identifies conserved pathway enrichment related to antiviral defense mechanisms, positive regulation of cytokine production, and regulation of response to biotic stimulus in APOBEC3A high monocytes. However, direct functional evidence for their mechanistic role in enhancing IC response is limited. Including in vitro or ex vivo functional assays (e.g., cytokine production, co-culture, or tumor cell killing assays) would provide much-needed biological context. If such experiments are not feasible at this stage, a more detailed discussion on the potential functions and clinical implications of these monocytes would be valuable.
3. For all cell-cell communication heatmaps (in Figure 1I, Figure 2G, Figure 3E, and supplemental Figures S3A, S6B, S7D2), clearly define what "Relative values" represent in the figure legends.
4. Beyond APOBEC3A expression, the manuscript also found that higher proportions of peripheral blood monocytes and, more specifically, classical monocytes within native tumors before treatment were independently and positively correlated with better IC outcomes. This raises an important question: is APOBEC3A expression truly necessary for predictive power, or could simpler monocyte quantification suffice? The authors should address whether APOBEC3A^{high} monocytes provide additive or independent predictive value beyond total/classical monocyte levels, ideally with supporting data or a stratified analysis. Additionally, the role of classical monocytes in the tumor microenvironment should be further discussed in the context of response modulation.

Minor Comments

1. The legend for Figure 2E states: "Volcano plot of DEGs with pCR/MPR versus NMMPR classical monocytes in native blood", which contradicts the figure title and text referring to "native tumors." Please revise the legend to reflect "pCR/MPR versus NMMPR classical monocytes in native tumors" for consistency.
2. The color coding of clusters differs between figure 5A and 5C, making visual comparison unnecessarily difficult. Please standardize the color schemes across related panels to improve readability.
3. In Figure 5C, APOBEC3A high monocytes belongs to which clusters of monocytes of healthy individuals and LUSC patients.
4. The manuscript uses terms like "APOBEC3A high" and "APOBEC3A +" somewhat interchangeably, which could be confusing. If they denote distinct phenotypes or thresholds, please define them clearly. Otherwise, consider unifying the terminology throughout the text for clarity.

Referee #2 (Comments on Novelty/Model System for Author):

Without in vitro and vivo validation, it remains difficult to establish the functional relevance and clinical importance of APOBEC3A^{high} monocytes in mediating response to immunochemotherapy. While the observed correlations are suggestive, mechanistic insights are lacking. The authors should consider alternative experimental systems and/or clinical datasets to support the causal role of this monocyte population, especially for predicting therapeutic response.

Referee #2 (Remarks for Author):

The study by Zhao et al. identifies APOBEC3A^{high} monocytes in peripheral blood and tumors as a potential biomarker for predicting immunochemotherapy (IC) response in patients with lung squamous cell carcinoma (LUSC). While the findings are intriguing and may have clinical implications, similar observations have already been reported—for example, in esophageal adenocarcinoma, where tumor monocyte content (TMC) was shown to predict IC outcomes (DOI: 10.1016/j.ccell.2023.06.006). Furthermore, the small patient cohort in the present study is inadequate for validating APOBEC3A^{high} monocytes as a biomarker, and no power analysis is provided to justify the conclusions. The term "biomarker" carries significant weight and should not be used without rigorous evidence. The authors tend to overstate the impact of their findings throughout the manuscript. To enhance the robustness, translational relevance, and novelty of the study, several methodological, technical, and interpretive concerns need to be addressed.

Major Concerns:

- The study includes only 15 patients, which significantly limits statistical power for biomarker discovery. To enhance the validity of the findings, the authors should expand the patient cohort and validate the results in an independent dataset.
- The detection of APOBEC3A^{high} monocytes in healthy donors is intriguing but lacks sufficient context. The authors should compare the phenotype and function of these cells between healthy individuals and LUSC patients. Importantly, circulating monocyte populations are known to be influenced by systemic inflammatory conditions. Therefore, to ensure the specificity of their findings for predicting IC response in LUSC, the authors should also include comparisons with patients experiencing non-malignant lung inflammation. This would help determine whether the observed APOBEC3A^{high} monocyte signature is truly disease-specific rather than a general marker of inflammation.
- The identification of rare monocyte populations using scRNA-seq may suffer from resolution limitations. Independent validation via flow cytometry or multiplex immunofluorescence is recommended.
- The shift in APOBEC3A^{high} monocytes post-IC is described, but its biological relevance is not established. Longitudinal tracking and/or functional assays (e.g., in vitro stimulation models) could clarify whether these changes reflect therapeutic impact or biological adaptation.
- While APOBEC3A^{high} monocytes are associated with improved IC response, their mechanistic contribution remains unexplored. Functional assays—such as monocyte depletion, adoptive transfer, or co-culture experiments—are needed to establish causality.

- Although GSEA highlights defense-related pathways, the mechanistic relevance to IC response is insufficiently explored. The authors should consider upstream regulator analysis (e.g., using Ingenuity Pathway Analysis) to identify key drivers of the observed transcriptional programs.
 - The presence of hyperexpanded TCR clones in NMPPR patients is noted, but their specificity remains unclear. Antigen-screening approaches should be used to determine whether these clones are tumor-reactive or bystander populations.
- Minor Concerns:
- The current manuscript title includes non-standard shorthand. The authors should revise the title to improve clarity and accessibility to a broad scientific and clinical audience.
 - Important clinical variables-such as smoking status, prior treatments, and comorbidities-were not accounted for in the analysis, which may introduce confounding effects. The authors should perform multivariate analyses to adjust for these potential confounders. Additionally, a table summarizing the clinicopathological characteristics of the patient cohort should be provided to enhance transparency and interpretability of the findings.
 - Although multiple samples and sequencing batches were integrated, the effectiveness of batch correction is not shown. The authors should provide supporting metrics (e.g., PCA or UMAP plots before and after integration) to demonstrate removal of batch effects.
 - DoubletFinder was used, but the criteria for doublet exclusion are not specified. The doublet rate and the rationale for the threshold should be reported to ensure transparency in monocyte purity.
 - Given the limitations of low-depth scRNA-seq data, CNV-based inference of tumor cells may lack resolution. The authors should validate tumor cell annotations using orthogonal methods such as immunohistochemistry for established tumor markers.
 - The CellChat analysis is informative but not experimentally verified. Key predicted ligand-receptor interactions should be tested via perturbation assays (e.g., blocking antibodies or recombinant proteins).
 - The study does not propose how APOBEC3A^{high} monocytes could be measured in a clinical setting. The authors should suggest feasible and scalable assays (e.g., flow cytometry or CyTOF) and comment on clinical translation potential, including cost and implementation logistics.
 - While raw sequencing data is deposited, processed data (e.g., cell type annotations, differential expression results) and analysis scripts are missing. The authors should provide processed data files and reproducible code via a public repository such as GitHub or Zenodo.

=====

As a service to authors, EMBO provides authors with the possibility to transfer a manuscript that one journal cannot offer to publish to another EMBO publication. The full manuscript and if applicable, reviewers reports are automatically sent to the receiving journal to allow for fast handling and a prompt decision on your manuscript. For more details of this service, and to transfer your manuscript to another EMBO title please click on Link Not Available

Dear Editors and Reviewers,

We sincerely appreciate the efforts of editors and reviewers in reviewing our manuscript entitled "APOBEC3A^{high} monocytes in native blood and tumors of LUSC patients predict better IC response" (Manuscript number: EMM-2025-22055). The paper's title has been changed to "**A Monocyte-centered Framework for Predicting Immunochemotherapy Efficacy in Lung Squamous Cell Carcinoma Patients**" based on our updated data. We have revised the manuscript accordingly. Below are our point-by-point responses to the reviewers' comments. Our revised version addressed the reviewers' concerns by using more patient samples to validate our conclusions. Hopefully, the revised version can be accepted for publication at *EMBO Molecular Medicine*.

First, we wish to highlight the clinical and scientific importance of this study. Lung cancer remains the leading cause of cancer-related mortality worldwide. Lung squamous cell carcinoma (LUSC), accounting for approximately 30% of lung cancer cases, still lacks effective targeted therapies. For advanced LUSC patients, immunochemotherapy represents the first-line treatment, yet reliable biomarkers for predicting its efficacy are extremely limited. Crucially, conducting multi-omics analyses on matched pre- and post-treatment samples from the same patient is exceptionally challenging due to practical difficulties in sample acquisition, patient compliance, and the need to preserve sample integrity. To our knowledge, no prior study has performed matched single-cell multi-omics profiling on both peripheral blood and primary tumor tissues from LUSC patients before and after immunochemotherapy. In this study, we successfully analyzed such paired samples from 15 patients (from an initial cohort of over 30, with others excluded due to sample degradation or quality issues). Therefore, we believe that, if accepted, our manuscript would represent the first report of its kind in this field.

Secondly, we have carefully addressed your constructive suggestions. As recommended, we have performed independent validation of our key single-cell multi-omics findings in expanded patient cohorts. These additional supporting data are now presented in Figure 1G, Figure 2D, and Figure 3E, strengthening the robustness and translational potential of our monocyte-centered predictive framework.

Specifically, to validate our previous single-cell omics findings, we further validated these results using more patients, such as native peripheral blood complete blood count (CBC) data from 228 patients, to confirm that the peripheral blood monocyte ratio was significantly higher in LUSC patients with a pathologic complete response (pCR) or major pathologic response (MPR) than in those without MPR (Figure 1G; Rebuttal Figure 1). We further validated that the proportion of classical monocytes among primary tumor monocytes was markedly higher in 34 pCR/MPR LUSC patients than in 20 non-MPR patients using multiplex immunofluorescence (IF) staining (Figures 2D; Rebuttal Figure 2). Moreover, the ratio of APOBEC3A⁺ monocytes among total monocytes in native primary tumors was significantly higher in the additional validation using multiplex IF staining analyses of 34 pCR/MPR LUSC cases compared with the 20 non-MPR ones (Figures 3E; Rebuttal Figure 3). We also identified thresholds of three indicators for clinical applications (Table S8).

Last, regarding the insightful suggestion to further investigate the functional role of monocytes in

driving immunochemotherapy resistance in LUSC, we agree this is a compelling direction for future research. However, a definitive functional validation would require sophisticated *in vivo* models (e.g., immunocompetent mouse models of LUSC) and ideally, validation in a clinical trial setting, which extends beyond the current scope and focus of this manuscript. Our present work aims primarily to establish a novel predictive framework based on comprehensive clinical multi-omics data.

Once again, we are deeply grateful for the considerable time and expertise you have dedicated to reviewing our manuscript and for providing suggestions that have significantly enhanced its quality. We have thoroughly enjoyed this constructive revision process.

We wish you all a joyful holiday season and a happy New Year.

Sincerely,

Kai & Jian

Reviewer#1's Comments:

The manuscript presents compelling novel findings by identifying APOBEC3A high monocytes in both native peripheral blood and primary tumors as a robust biomarker predicting better immunochemotherapy (IC) response in lung squamous cell carcinoma (LUSC) patients. This directly addresses a critical unmet clinical need for reliable predictive biomarkers in LUSC, where current approaches (e.g., PD-L1, TMB) have limited accuracy due to tumor heterogeneity. The use of multi-omics (scRNA-seq and scTCR-seq) on paired pre- and post-treatment samples from both peripheral blood and primary tumors in LUSC patients is methodologically rigorous and provides significant depth of insight into immune cell dynamics and biomarker discovery.

Overall, I consider this work to be a valuable contribution to the field. However, to enhance its impact and ensure the robustness of its conclusions, several key areas require clarification and further validation.

Major Comments

1. The study's findings are based on a cohort of 15 LUSC patients. While the single-cell depth is high, this is a relatively small number for biomarker validation, especially given the observed heterogeneity. Additional validation in larger, independent cohorts—either through scRNA-seq or targeted bulk approaches—would significantly strengthen the generalizability and clinical relevance of the findings.

Response: We thanked Reviewer #1 for pointing out this issue.

Specifically, to validate our previous single-cell omics findings, we further validated these results using more patients, such as native peripheral blood complete blood count (CBC) data from 228 patients, to confirm that the peripheral blood monocyte ratio was significantly higher in LUSC patients with a pathologic complete response (pCR) or major pathologic response (MPR) than in those without MPR (Figure 1G; Rebuttal Figure 1). We further validated that the proportion of classical monocytes among primary tumor monocytes was markedly higher in 34 pCR/MPR LUSC patients than in 20 non-MPR patients using multiplex immunofluorescence (IF) staining (Figures 2D; Rebuttal Figure 2). Moreover, the ratio of APOBEC3A⁺ monocytes among total monocytes in native primary tumors was significantly higher in the additional validation using multiplex IF staining analyses of 34 pCR/MPR LUSC cases compared with the 20 non-MPR ones (Figures 3E; Rebuttal Figure 3). We also identified thresholds of three indicators for clinical applications (Table S8).

Figure 1G; Rebuttal figure 1. Box plot and statistical comparison of monocyte complete blood count (CBC) ratios (without granulocytes) between pCR/MPR and NMPR patients.

Figures 2D; Rebuttal figure 2. Multiplex IF staining and statistical comparison of CD14 (purple), CD16 (red), and DAPI (blue) in pre-treatment LUSC tumor sections from patients with different responsiveness; scale bars: zoomed-in 10 μ m; overview 40 μ m.

Figures 3E; Rebuttal Figure 3. Multiplex IF staining and statistical comparison of CD14 (purple), CD16 (red), and APOBEC3A (green) in pre-treatment LUSC tumor sections from patients with varying residual tumor loads; scale bars: zoomed-in 10 μ m; overview 40 μ m.

- The study successfully identifies conserved pathway enrichment related to antiviral defense mechanisms, positive regulation of cytokine production, and regulation of response to biotic stimulus in APONEC3A high monocytes. However, direct functional evidence for their mechanistic role in enhancing IC response is limited. Including in vitro or ex vivo functional assays (e.g., cytokine production, co-culture, or tumor cell killing assays) would provide much-needed biological context. If such experiments are not

feasible at this stage, a more detailed discussion on the potential functions and clinical implications of these monocytes would be valuable.

Response: We thanked Reviewer #1 for pointing out this issue. As described at the beginning of this response letter, the main point of our paper was to identify the indicators predicting IC response of LUSC patients. The current manuscript did not aim to investigate the function of APOBEC3A^{high} monocytes in IC treatment. Therefore, we are planning to explore the function of monocytes in the future studies.

According to the suggestion of Reviewer#1, we have added a more detailed discussion on the potential functions and clinical implications of these monocytes.

The Potential Functions of these monocytes (Lines 262-271 in the Revised Manuscript):

“The role of monocytes in shaping IC efficacy might be explained by their plasticity in differentiating into macrophages or dendritic cells, thus influencing antigen presentation, immune activation, and T-cell fitness within the LUSC microenvironment.^{14,15,48} Notably, APOBEC3A⁺ monocytes were enriched in responders, suggesting a functional link with APOBEC-associated genomic instability, which has previously been associated with enhanced immunogenicity and improved responses to immunotherapy in multiple cancers.²⁹⁻³² Our SCENIC-based regulatory network analysis further identified IRF7 as a putative upstream regulator of APOBEC3A in monocytes, consistent with IRF-mediated transcriptional programs in antiviral responses and type-I interferon signaling (Figure S8D-F).^{49,50} These data collectively implied that APOBEC3A⁺ monocytes might not simply serve as correlative indicators, but could participate in remodeling immune landscapes during IC treatment, warranting deeper functional investigation.”

The Potential Functions of these monocytes (Lines 287-297 in the Revised Manuscript):

“Limitations of the Study

Although APOBEC3A⁺ monocytes correlated with IC response, their exact functional roles remained undefined. Single-cell transcriptomic analyses alone could not confirm whether APOBEC3A contributed directly to enhanced antitumor immunity or reflected an antiviral-like activation state driven by interferon responses.⁵⁴ In addition, interactions between this monocyte subset and effector immune cells were not fully resolved by computational analyses, and future studies employing lineage tracing or spatial multi-omics will be necessary to clarify their functional relevance. Moreover, whether APOBEC3A⁺ monocytes represent a tumor-induced phenotype or a physiologic subset requires comparison with monocytes from healthy donors. Investigations using matched control cohorts will be essential to determine whether these cells constitute an adaptation to tumor-driven inflammatory cues or reflect broader immunological programming.”

The clinical implications of these monocytes (Lines 245-260 in the Revised Manuscript):

“For advanced LUSC patients, immunochemotherapy represents the first-line treatment, yet reliable biomarkers for predicting its efficacy are extremely limited. Crucially, conducting multi-omics analyses on matched pre- and post-treatment samples from the same patient is quite challenging. To our knowledge, no prior study has performed matched single-cell multi-omics profiling on both peripheral blood and primary tumor tissues from LUSC patients before and after immunochemotherapy. Here, we used single-cell multi-omics profiling of paired pre- and post-treatment tumor and blood samples to demonstrate that elevated levels of three monocyte-derived indicators, including native circulating monocytes, native tumor classical monocytes, and APOBEC3A+ monocytes across both compartments, were strongly associated with pathologic responses to immunochemotherapy (IC) in patients with LUSC, validated by more independent patients’ samples. These findings advanced prior work implicating monocytes as key modulators of tumor-immune interactions across cancers⁴⁵⁻⁴⁷ and extended current biomarker research beyond tumor-restricted measurements such as PD-L1 or TMB, which remain limited by spatial heterogeneity and sampling variability in clinical settings⁷⁻⁹. By integrating paired pre- and post-treatment samples from peripheral blood and tumors, our findings supported a systemic-intratumoral axis through which monocyte composition reflects IC responsiveness in LUSC patients, offering a clinically accessible strategy for LUSC patient selection and monitoring.”

The clinical implications of these monocytes (Lines 281-285 in the Revised Manuscript):

“Together, our findings underscore the translational potential of monocyte-focused assessment as a minimally invasive approach to guide treatment stratification, capturing immunological states at the intersection of systemic and local tumor environments. This paired-sampling strategy, rarely applied in previous IC biomarker studies, may accelerate predictive indicator discovery and deepen mechanistic understanding of IC responsiveness in LUSC.”

3. For all cell-cell communication heatmaps (in Figure 1I, Figure 2G, Figure 3E, and supplemental Figures S3A, S6B, S7D2), clearly define what “Relative values” represent in the figure legends.

Response: We thanked Reviewer #1 for pointing out this issue.

We have added a clear definition of “Relative values” in the Method section. The “Relative values” represent the cell-cell interaction strength value between monocytes and other cells in pCR/MPR patients relative to NMPR patients. “Positive Relative values” means the cell-cell interaction strength value between monocytes and other cells in pCR/MPR patients is higher than that in NMPR patients, and vice versa.

Since the current manuscript did not aim to investigate the function of monocytes and their subpopulations in IC treatment, the results of cell-cell communication have been moved into supplement figures, and the panel labels of **previous Figures 1I, 2G, and 3E** have been adjusted into **Figures S3A, S6C, and S8A (Rebuttal Figure 4)**. Additionally, the panel labels of **previous Figures S3A, S6B, and S7D2** have been adjusted into **Figure S3B, S6D, S6E, and S8B**.

Figure S3A, Figure S6C, Figure S8A (previous labeled as Figure 1I, Figure 2G, Figure3E); Rebuttal Figure 4. Cell-cell communication difference heatmap among monocytes/classical monocytes/APOBEC3A+ monocytes, and other major cell types with pCR/MPR versus NMPR.

4. Beyond APOBEC3A expression, the manuscript also found that higher proportions of peripheral blood monocytes and, more specifically, classical monocytes within native tumors before treatment were independently and positively correlated with better IC outcomes. This raises an important question: is APOBEC3A expression truly necessary for predictive power, or could simpler monocyte quantification suffice? The authors should address whether APOBEC3A^{high} monocytes provide additive or independent predictive value beyond total/classical monocyte levels, ideally with supporting data or a stratified analysis. Additionally, the role of classical monocytes in the tumor microenvironment should be further discussed in the context of response modulation.

Response: We thanked Reviewer #1 for pointing out this issue. We agreed with the suggestions and have changed the main conclusion and title of the paper.

APOBEC3A^{high} monocytes in native primary LUSC tumors provide an independent predictive value beyond classical monocyte levels in native primary LUSC tumors, considering that APOBEC3A^{high} monocytes in native peripheral blood also provide a positive predictive value of IC response in LUSC patients.

Now we claimed three strategies identified to predict the IC response of LUSC patients, including total monocyte levels in native peripheral blood complete blood count (CBC), classical monocyte levels in native primary tumors, and APOBEC3A⁺ monocytes (here we have unified APOBEC3A^{high} to APOBEC3A⁺, sorry for previous confusion) in native peripheral blood and primary tumors. The title of the paper has been changed into “**A Monocyte-centered Framework for Predicting Immunotherapy Efficacy in Lung Squamous Cell Carcinoma Patients**”.

According to our revised conclusions and title, we have validated the predictive value of total/classical monocyte levels by comparing the native peripheral blood CBC test ratio of monocytes, and comparing classical monocyte ratio based on IF staining between pCR/MPR and NMPR patients (Figures 1G and 2D; Rebuttal Figures 1 and 2).

Moreover, we discussed the role of classical monocytes in the tumor microenvironment in the revised manuscript, listed below (Lines 138-158 in the Revised Manuscript).

“To further investigate the differences of classical monocytes in primary tumors between pCR/MPR and NMPR patients before treatment, we identified DEGs (e.g., APOBEC3A, IFI6) and conducted GSEA. Compared to NMPR, classical monocytes in native pCR/MPR tumors showed enrichment in pathways such as responses to the virus, responses to cytokine, cell surface receptor signaling, cellular response to cytokine stimulus, and viral genome replication (Figures S6A and S6B).

Similar to the interaction patterns observed in monocytes (Figure S3A), classical monocytes from pCR/MPR patients exhibited weaker interactions with most other cell types (e.g., T cells) in native blood compared to NMPR patients (Figure S6C). Conversely, within native tumor samples, these results suggested that these interactions were stronger in pCR/MPR patients than in NMPR patients (Figure S6C).

IC treatment enhanced the interaction strength of classical and intermediate monocytes as receptors with most cell types—including intermediate monocytes, T cells, macrophages, DC cells, mast cells, and NK cells—in blood samples of pCR/MPR patients compared to NMPR patients (Figures S6C and S6D). In contrast, within tumors, IC treatment reduced the interaction strength of classical and intermediate monocytes as receptors with various cell types—including all three monocyte subtypes, macrophages, fibroblasts, normal epithelial cells, endothelial cells, DC cells, mast cells, B cells, and NK cells—in pCR/MPR compared to NMPR patients (Figures S6C and S6E). In summary, classical monocytes in native tumors from pCR/MPR patients showed potentially stronger interactions with other cell types (e.g., T cells) than those in tumors from NMPR patients, except with tumor cells, plasma cells, and ciliated cells.”

Minor Comments

- 1. The legend for Figure 2E states: “Volcano plot of DEGs with pCR/MPR versus NMPR classical monocytes in native blood”, which contradicts the figure title and text referring to “native tumors.” Please revise the legend to reflect “pCR/MPR versus NMPR classical monocytes in native tumors” for consistency.**

Response: We thanked Reviewer #1 for pointing out this issue and apologized for any confusion. We have corrected the figure legend and shown it as below (Figure S6A; Rebuttal Figure 5). The panel labels in **Figure 2E** of the previous version have been moved to **Figure S6A** (Rebuttal Figure 5) in the revised version.

Figure S6A; Rebuttal Figure 5. Volcano plot of DEGs with pCR/MPR versus NMMPR classical monocytes in native tumor; the x-axis indicates percentage differences in gene-expressing cells between groups, and the y-axis shows $-\log_{10}(\text{adjusted p-value})$. Dots are colored based on $\log_2\text{FC}$ values.

2. The color coding of clusters differs between figure 5A and 5C, making visual comparison unnecessarily difficult. Please standardize the color schemes across related panels to improve readability.

Response: We thanked Reviewer #1 for pointing out this issue and apologized for any confusion.

We have adapted the color scheme of Figure 5C (now removed from the main figures) to match that of Figure 5A (now removed from the main figures) for better readability (**Rebuttal Figures 6 and 7**).

Since the current main point of our paper is to identify indicators predicting IC response in LUSC patients, we removed the content of Figure 5A from the previous version, which mainly explored the potential origin of APOBEC3A+ monocytes.

Previous Figure 5A (Removed from the previous version); Rebuttal Figure 6. UMAP embedding of integrated monocyte populations, including healthy and LUSC patient-derived monocytes.

Previous Figure 5C (removed); Rebuttal Figure 7. Relative cluster proportions of clusters from Figure 5A in healthy and patient groups.

3. In Figure 5C, APOBEC3A high monocytes belongs to which clusters of monocytes of healthy individuals and LUSC patients.

Response: We thanked Reviewer #1 for pointing out this issue and apologized for any confusion.

We labeled Clusters 3, 5, 6, and 8 as APOBEC3A^{high} monocytes and other clusters as APOBEC3A^{low} monocytes in Figure 5C, which was removed in the revised manuscript due to that the current main point of our paper is to identify indicators predicting IC response in LUSC patients.

4. The manuscript uses terms like “APOBEC3A high” and “APOBEC3A+” somewhat interchangeably, which could be confusing. If they denote distinct phenotypes or thresholds, please define them clearly. Otherwise, consider unifying the terminology throughout the text for clarity.

Response: We thanked Reviewer #1 for pointing out this issue and apologized for any confusion.

We have unified “APOBEC3A high” to “APOBEC3A+” throughout the text and figures.

Reviewer#2

Reviewer#2's Comments:

Without *in vitro* and *in vivo* validation, it remains difficult to establish the functional relevance and clinical importance of APOBEC3A^{high} monocytes in mediating response to immunochemotherapy. While the observed correlations are suggestive, mechanistic insights are lacking. The authors should consider alternative experimental systems and/or clinical datasets to support the causal role of this monocyte population, especially for predicting therapeutic response.

The study by Zhao et al/ identifies APOBEC3A^{high} monocytes in peripheral blood and tumors as a potential biomarker for predicting immunochemotherapy (IC) response in patients with lung squamous cell carcinoma (LUSC). While the findings are intriguing and may have clinical implications, similar observations have already been reported—for example, in esophageal adenocarcinoma, where tumor monocyte content (TMC) was shown to predict IC outcomes (DOI: 10.1016/j.cell/2023.06.006). Furthermore, the small patient cohort in the present study is inadequate for validating APOBEC3A^{high} monocytes as a biomarker, and no power analysis is provided to justify the conclusions. The term “biomarker” carries significant weight and should not be used without rigorous evidence. The authors tend to overstate the impact of their findings throughout the manuscript. To enhance the robustness, translational relevance, and novelty of the study, several methodological, technical, and interpretive concerns need to be addressed.

Response: We sincerely appreciate the efforts of reviewers in reviewing our manuscript entitled "APOBEC3A^{high} monocytes in native blood and tumors of LUSC patients predict better IC response" (Manuscript number: EMM-2025-22055). The title of the paper has been changed into “**A Monocyte-centered Framework for Predicting Immunochemotherapy Efficacy in Lung Squamous Cell Carcinoma Patients**” based on our updated data. We have revised the manuscript accordingly. Below are our point-by-point responses to the reviewers' comments. Our revised version addressed the reviewers' concerns with the updated results using more patient samples to validate our conclusions.

First, we wish to highlight the clinical and scientific importance of this study. Lung cancer remains the leading cause of cancer-related mortality worldwide. Lung squamous cell carcinoma (LUSC), accounting for approximately 30% of lung cancer cases, still lacks effective targeted therapies. For advanced LUSC patients, immunochemotherapy represents the first-line treatment, yet reliable biomarkers for predicting its efficacy are extremely limited. Crucially, conducting multi-omics analyses on matched pre- and post-treatment samples from the same patient is exceptionally challenging due to practical difficulties in sample acquisition, patient compliance, and the need to preserve sample integrity. To our knowledge, no prior study has performed matched single-cell multi-omics profiling on both peripheral blood and primary tumor tissues from LUSC patients before and after immunochemotherapy. In this study, we successfully analyzed such paired samples from 15 patients (from an initial cohort of over 30, with others excluded due to sample degradation or quality issues). Therefore, we believe that, if accepted, our manuscript would represent the first report of its kind in this field.

Secondly, we have carefully addressed your constructive suggestions. As recommended, we have performed independent validation of our key single-cell multi-omics findings in expanded patient cohorts. These additional supporting data are now presented in Figure 1G, Figure 2D, and Figure 3E, strengthening the robustness and translational potential of our monocyte-centered predictive framework.

Specifically, to validate our previous single-cell omics findings, we further validated these results using more patients, such as native peripheral blood complete blood count (CBC) data from 228 patients, to confirm that the peripheral blood monocyte ratio was significantly higher in LUSC patients with a pathologic complete response (pCR) or major pathologic response (MPR) than in those without MPR (Figure 1G; Rebuttal Figure 1). We further validated that the proportion of classical monocytes among primary tumor monocytes was markedly higher in 34 pCR/MPR LUSC patients than in 20 non-MPR patients using multiplex immunofluorescence (IF) staining (Figures 2D; Rebuttal Figure 2). Moreover, the ratio of APOBEC3A⁺ monocytes among total monocytes in native primary tumors was significantly higher in the additional validation using multiplex IF staining analyses of 34 pCR/MPR LUSC cases compared with the 20 non-MPR ones (Figures 3E; Rebuttal Figure 3). We also identified thresholds of three indicators for clinical applications (Table S8).

Last, regarding the insightful suggestion to further investigate the functional role of monocytes in driving immunochemotherapy resistance in LUSC, we agree this is a compelling direction for future research. However, a definitive functional validation would require sophisticated *in vivo* models (e.g., immunocompetent mouse models of LUSC) and ideally, validation in a clinical trial setting, which extends beyond the current scope and focus of this manuscript. Our present work aims primarily to establish a novel predictive framework based on comprehensive clinical multi-omics data.

Once again, we are deeply grateful for the considerable time and expertise you have dedicated to reviewing our manuscript and for providing suggestions that have significantly enhanced its quality. We have thoroughly enjoyed this constructive revision process.

We wish you all a joyful holiday season and a happy New Year.

Sincerely,

Jian

Major Concerns:

- 1. The study includes only 15 patients, which significantly limits statistical power for biomarker discovery. To enhance the validity of the findings, the authors should expand the patient cohort and validate the results in an independent dataset.**

Response: We thanked Reviewer #2 for pointing out this issue.

Per the Reviewer's suggestion, to validate our previous single-cell omics findings, we further validated these results using more patients, such as native peripheral blood complete blood count

(CBC) data from 228 patients, to confirm that the peripheral blood monocyte ratio was significantly higher in LUSC patients with a pathologic complete response (pCR) or major pathologic response (MPR) than in those without MPR (Figure 1G; Rebuttal Figure 1). We further validated that the proportion of classical monocytes among primary tumor monocytes was markedly higher in 34 pCR/MPR LUSC patients than in 20 non-MPR patients using multiplex immunofluorescence (IF) staining (Figures 2D; Rebuttal Figure 2). Moreover, the ratio of APOBEC3A⁺ monocytes among total monocytes in native primary tumors was significantly higher in the additional validation using multiplex IF staining analyses of 34 pCR/MPR LUSC cases compared with the 20 non-MPR ones (Figures 3E; Rebuttal Figure 3). We also identified thresholds of three indicators for clinical applications (Table S8).

Moreover, we wish to highlight the clinical and scientific importance of this study. Lung cancer remains the leading cause of cancer-related mortality worldwide. Lung squamous cell carcinoma (LUSC), accounting for approximately 30% of lung cancer cases, still lacks effective targeted therapies. For advanced LUSC patients, immunochemotherapy represents the first-line treatment, yet reliable biomarkers for predicting its efficacy are extremely limited. Crucially, conducting multi-omics analyses on matched pre- and post-treatment samples from the same patient is exceptionally challenging due to practical difficulties in sample acquisition, patient compliance, and the need to preserve sample integrity. To our knowledge, no prior study has performed matched single-cell multi-omics profiling on both peripheral blood and primary tumor tissues from LUSC patients before and after immunochemotherapy. In this study, we successfully analyzed such paired samples from 15 patients (from an initial cohort of over 30, with others excluded due to sample degradation or quality issues). Therefore, we believe that, if accepted, our manuscript would represent the first report of its kind in this field.

Figure 1G; Rebuttal figure 1. Box plot and comparisons of monocyte complete blood count (CBC) ratios (without granulocytes) between pCR/MPR and NMPR patients.

Figures 2D; Rebuttal figure 2. Multiplex IF staining and statistical comparison of CD14 (purple),

CD16 (red), and DAPI (blue) in pre-treatment LUSC tumor sections from patients with different responsiveness; scale bars: zoomed-in 10 μ m; overview 40 μ m.

Figures 3E; Rebuttal Figure 3. Multiplex IF staining and statistical comparison of CD14 (purple), CD16 (red), and APOBEC3A (green) in pre-treatment LUSC tumor sections from patients with varying residual tumor loads; scale bars: zoomed-in 10 μ m; overview 40 μ m.

2. The detection of APOBEC3A^{high} monocytes in healthy donors is intriguing but lacks sufficient context. The authors should compare the phenotype and function of these cells between healthy individuals and LUSC patients. Importantly, circulating monocyte populations are known to be influenced by systemic inflammatory conditions. Therefore, to ensure the specificity of their findings for predicting IC response in LUSC, the authors should also include comparisons with patients experiencing non-malignant lung inflammation. This would help determine whether the observed APOBEC3A^{high} monocyte signature is truly disease-specific rather than a general marker of inflammation.

Response: We thanked Reviewer #2 for pointing out this issue.

We agreed with the reviewer that circulating monocyte populations were known to be influenced by systemic inflammatory conditions. Therefore, we didn't exclude the possibility that APOBEC3A⁺ monocyte signature might be a general marker of inflammation. Actually, this didn't affect our current conclusion that APOBEC3A⁺ monocytes positively predicted the IC response of LUSC patients.

The main point of our paper was to identify the indicators to predict the IC response of LUSC patients. Therefore, we further validated that the ratio of APOBEC3A⁺ monocytes among total monocytes in native primary tumors was significantly higher in the additional validation using multiplex IF staining analyses of 34 pCR/MPR LUSC cases compared with the 20 non-MPR ones (Figures 3E; Rebuttal Figure 3). Meanwhile, we didn't exclude the possibility of APOBEC3A⁺ monocytes associated with other diseases.

3. The identification of rare monocyte populations using scRNA-seq may suffer from resolution limitations. Independent validation via flow cytometry or multiplex immunofluorescence is recommended.

Response: We thanked Reviewer #2 for pointing out this issue.

The multiplex immunofluorescence results of native tumors were shown in Figure 2D (Rebuttal Figure 2) and Figure 3E. We have also enlarged the size of multiplex immunofluorescence validation in native tumors up to 54 patients. For blood monocyte ratio validation, we collected 228 complete native peripheral blood complete blood count (CBC) test data from LUSC patients before IC and confirmed a significant increase in blood monocyte ratios in pCR/MPR patients compared to NMPR patients (Figure 1G; Rebuttal Figure 1).

4. The shift in APOBEC3A^{high} monocytes post-IC is described, but its biological relevance is not established. Longitudinal tracking and/or functional assays (e.g., in vitro stimulation models) could clarify whether these changes reflect therapeutic impact or biological adaptation.

Response: We thanked Reviewer #2 for pointing out this issue.

Regarding the insightful suggestion to further investigate the functional role of APOBEC3A^{high} monocytes in regulating immunochemotherapy resistance in LUSC, we agree this is a compelling direction for future research. However, a definitive functional validation would require sophisticated *in vivo* models (e.g., immunocompetent mouse models of LUSC) and ideally, validation in a clinical trial setting, which extends beyond the current scope and focus of this manuscript. Our present work aims primarily to establish a novel predictive framework based on comprehensive clinical multi-omics data. Therefore, longitudinal tracking and/or functional assays of APOBEC3A^{high} monocytes during IC were planned for the next project, not this manuscript. The main point of our paper was to identify the indicators predicting IC response of LUSC patients. Therefore, we concentrated on the difference among patients with different IC responses before treatment instead of after treatment.

5. While APOBEC3A^{high} monocytes are associated with improved IC response, their mechanistic contribution remains unexplored. Functional assays—such as monocyte depletion, adoptive transfer, or co-culture experiments—are needed to establish causality.

Response: We thanked Reviewer #2 for pointing out this issue.

Regarding the great suggestion to further investigate the functional role of APOBEC3A^{high} monocytes in regulating improved IC response in LUSC, we agree this is a compelling direction for future research. However, a definitive functional validation would require sophisticated *in vivo* models (e.g., immunocompetent mouse models of LUSC) to conduct monocyte depletion or adoptive transfer and ideally, validation in a clinical trial setting, which extends beyond the current scope and focus of this manuscript. Our present work aims primarily to establish a novel predictive

framework based on comprehensive clinical multi-omics data. Therefore, functional assays-such as monocyte depletion, adoptive transfer, or co-culture analyses of APOBEC3A^{high} monocytes during IC were planned for the next project, not this manuscript. Therefore, we concentrated on the difference among patients with different IC responses before treatment instead of after treatment in our current version.

- 6. Although GSEA highlights defense-related pathways, the mechanistic relevance to IC response is insufficiently explored. The authors should consider upstream regulator analysis (e.g., using Ingenuity Pathway Analysis) to identify key drivers of the observed transcriptional programs.**

Response: We thanked Reviewer #2 for pointing out this issue.

We analyzed the upstream regulator of APOBEC3A in monocytes by SCENIC (Single-Cell rEgulatory Network Inference and Clustering, detailed information in Method). By overlapping the potential upstream transcriptome regulators (TFs) of APOBEC3A in blood monocytes and tumor monocytes, we identified IRF7 as potential conserved upstream regulator of APOBEC3A in blood and tumor monocytes (Figure S8D; Rebuttal Figure 8). We also observed the co-expression of APOBEC3A and IRF7 regulation activity (IRF7(+)) in monocytes (Figures S8E and S8F; Rebuttal Figure 9). Additionally, we added the related content in the discussion section.

Figure S8D; Rebuttal Figure 8. Overlap of upstream TFs of APOBEC3A in native blood and tumor monocytes.

Figures S8E and S8F; Rebuttal Figure 9. Expression density of APOBEC3A; Co-expression density of APOBEC3A and IRF7(+) in native blood and tumor monocytes.

Discussion supplement (Lines 262-271 in the revised manuscript):

“The role of monocytes in shaping IC efficacy might be explained by their plasticity in differentiating into macrophages or dendritic cells, thus influencing antigen presentation, immune activation, and T-cell fitness within the LUSC microenvironment.^{14,15,48} Notably, APOBEC3A+ monocytes were enriched in responders, suggesting a functional link with APOBEC-associated genomic instability, which has previously been associated with enhanced immunogenicity and improved responses to immunotherapy in multiple cancers.²⁹⁻³² Our SCENIC-based regulatory network analysis further identified IRF7 as a putative upstream regulator of APOBEC3A in monocytes, consistent with IRF-mediated transcriptional programs in antiviral responses and type-I interferon signaling (Figure S8D-F).^{49,50} These data collectively implied that APOBEC3A+ monocytes might not simply serve as correlative indicators, but could participate in remodeling immune landscapes during IC treatment, warranting deeper functional investigation.”

- 7. The presence of hyperexpanded TCR clones in NMPR patients is noted, but their specificity remains unclear. Antigen-screening approaches should be used to determine whether these clones are tumor-reactive or bystander populations.**

Response: We thanked Reviewer #2 for pointing out this issue.

The main point of our paper was to identify indicators predicting IC response in LUSC patients, which we achieved by demonstrating that monocytes positively correlate with IC efficiency in treating LUSC. Now, the TCR differences between NMPR and pCR/MPR patients were not universal, and failure was an indicator. Therefore, we only showed TCR differences in LUSC patients before and after treatment, rather than further exploring the underlying mechanisms. To

avoid confusion about the main point, we have moved the TCR analysis results to the supplement figures.

Minor Concerns:

- 1. The current manuscript title includes non-standard shorthand. The authors should revise the title to improve clarity and accessibility to a broad scientific and clinical audience.**

Response: We thanked Reviewer #2 for pointing out this issue and apologized for any confusion. We have changed title into “**A Monocyte-centered Framework for Predicting Immunochemotherapy Efficacy in Lung Squamous Cell Carcinoma Patients**”.

- 2. Important clinical variables-such as smoking status, prior treatments, and comorbidities-were not accounted for in the analysis, which may introduce confounding effects. The authors should perform multivariate analyses to adjust for these potential confounders. Additionally, a table summarizing the clinicopathological characteristics of patient cohort should be provided to enhance transparency and interpretability of findings.**

Response: We thanked Reviewer #2 for pointing out this issue and apologize for any confusion. We performed multivariate analyses by comparing variation distributions between pCR/MPR and NMPR patients and found no significant difference (Figure S1A; Rebuttal Figure 10, Table S2). Additionally, the table containing the clinical information and clinicopathological characteristics of samples for validation has also been added (Table S2, S4-S5).

Figure S1A; Rebuttal Figure 10. Box plot and comparisons of smoking index between single-cell sequenced pCR/MPR and NMPR patients.

Characteristic	Total Patients (N=15)	Responders (n=8)	Non-responders (n=7)
Age, years			
Median (Range)	63 (53-72)	62 (53-72)	67 (56-72)
Gender, n (%)			
Male	15 (100.0)	8 (100.0)	7 (100.0)
Female	0	0	0
Pathological Type, n (%)			
Lung Squamous Cell Carcinoma (LUSC)	15 (100.0)	8 (100.0)	7 (100.0)
Smoking Status, n (%)			
Smoker	14 (93.3)	7 (87.5)	7 (100.0)
Non-smoker	1 (6.7)	1(12.5)	0
Tumor Stage, n (%)			
IIB	3 (20.0)	1(12.5)	2 (28.6)
IIIA	5 (33.3)	4 (50.0)	1 (14.3)
IIIB	7 (46.7)	3 (37.5)	4 (57.1)
PD-L1 Expression, n (%)			
<1%	1 (6.7)	0	1 (14.3)
1-50%	3 (20.0)	1 (12.5)	2 (28.6)
>50%	2 (13.3)	2 (25.0)	0
Unknown	9 (60.0)	5 (62.5)	4 (57.1)
Treatment Cycles, n (%)			
2 cycles	12 (80.0)	7 (87.5)	5 (71.4)
3 cycles	3 (20.0)	1 (12.5)	2 (28.6)
Treatment Regimen, n (%)			
Nab-paclitaxel+Carboplatin+Pembrolizumab	1 (6.7)	1 (12.5)	0
Nab-paclitaxel+Carboplatin+Sintilimab	7 (46.7)	4 (50.0)	3 (42.9)
Nab-paclitaxel+Carboplatin+Toripalimab	6 (40.0)	3 (37.5)	3 (42.9)
Nab-paclitaxel+Cisplatin+Sintilimab	1(6.7)	0	1(14.3)

Table S2. The clinical characteristic distribution of scRNA-seq cohort.

3. Although multiple samples and sequencing batches were integrated, the effectiveness of batch correction is not shown. The authors should provide supporting metrics (e.g., PCA or UMAP plots before and after integration) to demonstrate removal of batch effects.

Response: The reviewer's guidance was very important.

Accordingly, we have added UMAP plots before and after integration to demonstrate the removal of batch effects (Figure S1B; Rebuttal Figure 11). The UMAP showed that monocytes in peripheral blood and primary tumors were not in the same place before removing batch effects. Additionally, the tumor cells were also separated into plural clusters instead of one position of UMAP before removing batch effects. The discrete distribution of monocytes and tumor cells indicated the existence of batch effects in the raw scRNA-seq data, and confirmed the effectiveness of batch

correction compared to Figure 1C.

Figure S1B; Rebuttal Figure 11. UMAP of scRNA-seq data before removing batch effect

4. Doubletfinder was used, but the criteria for doublet exclusion are not specified. The doublet rate and the rationale for the threshold should be reported to ensure transparency in monocyte purity.

Response: The reviewer’s guidance was essential to us in improving the quality of our analyses. Accordingly, we have added the doublet rate and rationale for the threshold in the Method section (Lines 734-737 in the revised manuscript).

“The doublet rate was set 7.5%. According to the pipeline and doublet rate estimation table provided by the author of DoubletFinder, 10000 recovered cells corresponding to 7.5~7.6% doublet rate (Table S3; Rebuttal Figure 12). Because the recovered cell numbers of most scRNA-seq samples fluctuated nearby 10000 cells, we set 7.5% as the doublet rate.”

Multiplet Ratio (%)	of Cells Recovered
~0.4%	~500
~0.8%	~1000
~1.6%	~2000
~2.3%	~3000
~3.1%	~4000
~3.9%	~5000
~4.6%	~6000
~5.4%	~7000
~6.1%	~8000
~6.9%	~9000
~7.6%	~10000

Table S3; Rebuttal Figure 12. Doublet rate estimation table of *Doubletfinder* (referring to <https://github.com/chris-mcginnis-ucsf/DoubletFinder/issues/76>).

5. Given the limitations of low-depth scRNA-seq data, CNV-based inference of tumor cells

may lack resolution. The authors should validate tumor cell annotations using orthogonal methods such as immunohistochemistry for established tumor markers.

Response: Thanks a lot. We have added IF staining results to validate the tumor cell existence in the native tumor slices of single-cell sequenced samples (Figure S1E; Rebuttal Figure 13).

Figure S1E; Rebuttal Figure 13. IF staining of KRT5 (green) and DAPI (blue) of native tumor slices from patients in scRNA-seq cohort.

6. The CellChat analysis is informative but not experimentally verified. Key predicted ligand-receptor interactions should be tested via perturbation assays (e.g., blocking antibodies or recombinant proteins).

Response: The suggestions were helpful for our future research direction in investigating the roles of monocytes in IC treatment for LUSC patients. However, the main focus of our paper is to identify indicators predicting IC response in LUSC patients. Therefore, we plan to verify the ligand-receptor interactions and have moved the results of CellChat analysis into supplemental figures.

7. The study does not propose how APOBEC3A^{high} monocytes could be measured in a clinical setting. The authors should suggest feasible and scalable assays (e.g., flow cytometry or CyTOF) and comment on clinical translation potential, including cost and implementation logistics.

Response: We thanked Reviewer #2 for pointing out this issue and apologized for any confusion.

The feasible and scalable assays to measure APOBEC3A^{high} monocytes in the primary tumor was multiplex IF staining. Moreover, we found that the ratio of APOBEC3A⁺ monocytes among total monocytes in native primary tumors was significantly higher in an additional validation using multiplex IF staining analyses of 34 pCR/MPR LUSC cases compared with 20 non-MPR cases (Figures 3E; Rebuttal Figure 3).

To enhance the clinical utility of the three identified indicators, we designed an application strategy that balances prediction accuracy and coverage by iterating through thresholds and testing combinations of the three indicators (Table S8). In summary, doctors have three categories to adapt: if taking only one indicator, CBC monocyte ratio was the category with highest coverage and highest accuracy under the single indicator; if taking two indicators, CBC monocyte ratio or CD14+

monocyte ratio combined with APOBEC3A+ monocyte ratio in tumor was the best combination with 100% accuracy, the same as when combining all three indicators (Figure 3H; Rebuttal Figure 14).

The Predictive Coverage and Accuracy Table of Three Indicators with Different Combinations in IF			
Combination of Indicators (pCR/MPR n=22, NMMPR n=11)		pCR/MPR Coverage	pCR/MPR Accuracy
Single indicators	CBC Monocyte Ratio (>23.96%)	72.73%	84.21%
	CD14+ Monocyte Ratio (>33.34%)	68.18%	75.00%
	APOBEC3A+ Monocyte Ratio (>15.87%)	50.00%	68.75%
Double Indicators	CBC Monocyte Ratio (>23.96%) & CD14+ Monocyte Ratio (>33.34%)	54.55%	85.71%
	CBC Monocyte Ratio (>23.96%) & APOBEC3A+ Monocyte Ratio (>15.87%)	36.36%	100.00%
	CD14+ Monocyte Ratio (>33.34%) & APOBEC3A+ Monocyte Ratio (>15.87%)	36.36%	100.00%
	Random Double Indicators or More	68.18%	83.33%
Triple Indicators	Triple Indicators	36.36%	100.00%

Table S8; Rebuttal Figure 14. The predictive coverage and accuracy table for pCR/MPR patients based on three indicators with different combinations. The combination of indicators and thresholds were calculated based on scRNA-seq data, and were validated based on the paired CBC test data and IF staining results of native tumor slices ($n_{pCR/MPR}=22$, $n_{NMMPR}=11$). The pCR/MPR coverage meant the ratio of the numbers of pCR/MPR patients passing thresholds versus the sum of pCR/MPR patients ($n=22$). The pCR/MPR accuracy meant the ratio of the numbers of pCR/MPR patients passing thresholds versus the sum of patients passing thresholds.

8. While raw sequencing data is deposited, processed data (e.g., cell type annotations, differential expression results) and analysis scripts are missing. The authors should provide processed data files and reproducible code via a public repository such as GitHub or Zenodo.

Response: We have uploaded processed data files and reproducible code on GitHub (<https://github.com/ZJE-Zijin/Monocyte-centrered-IC-predictor>).

13th Jan 2026

Dear Prof. Wang,

Thank you for your appeal on our previous decision and for submitting a revised manuscript to EMBO Molecular Medicine. Following editorial evaluation of the revisions, we sent the manuscript back to the initial referees.

As you will see below, although referee #1 is satisfied with the revised manuscript, referee #2 still has concerns about the lack of a power calculation to justify the sizes of the validation cohorts. We would therefore like to invite you to revise the manuscript further to address this remaining concern.

As EMBO Press usually only allows one round of revisions, please be aware that this will be your last opportunity to address the remaining issues. The revised manuscript will be reviewed again, and we cannot guarantee a positive outcome at this stage.

Should you find that the requested revisions are not feasible within the constraints outlined here and prefer, therefore, to submit your paper elsewhere, we would welcome a message to this effect.

I look forward to seeing a revised form of your manuscript as soon as possible. Use this link to login to the manuscript system and submit your revision: <https://embomolmed.msubmit.net/cgi-bin/main.plex>

Yours sincerely,

Lise Roth

When preparing your revised manuscript, please refer to our guidelines: <https://link.springer.com/journal/44321/submission-guidelines#cms-Revised-submissions>. We perform an initial quality control of all revised manuscripts before re-review; failure to include requested items will delay the evaluation of your revision.

We require:

2) Individual production quality figure files as .eps, .tif, .jpg (one file per figure). For guidance, download the 'Figure Guide PDF': <https://media.springernature.com/original/springer-cms/rest/v1/content/27825798/data/v1>.

3) A .docx formatted letter INCLUDING the reviewers' reports and your detailed point-by-point responses to their comments. As part of the EMBO Press transparent editorial process, the point-by-point response is part of the Review Process File (RPF), which will be published alongside your paper.

4) A complete author checklist, which you can download from our author guidelines. Please insert information in the checklist that is also reflected in the manuscript. The completed author checklist will also be part of the RPF.

6) It is mandatory to include a 'Data Availability' section after the Materials and Methods. Before submitting your revision, primary datasets produced in this study need to be deposited in an appropriate public database, and the accession numbers and database listed under 'Data Availability'. Please remember to provide a reviewer password if the datasets are not yet public.

7) For data quantification: please specify the name of the statistical test used to generate error bars and P values, the number (n) of independent experiments (specify technical or biological replicates) underlying each data point and the test used to calculate p-values in each figure legend. The figure legends should contain a basic description of n, P and the test applied. Graphs must include a description of the bars and the error bars (s.d., s.e.m.).

9) Our journal encourages inclusion of *data citations in the reference list* to directly cite datasets that were re-used and obtained from public databases. Data citations in the article text are distinct from normal bibliographical citations and should directly link to the database records from which the data can be accessed. In the main text, data citations are formatted as follows: "Data ref: Smith et al, 2001" or "Data ref: NCBI Sequence Read Archive PRJNA342805, 2017". In the Reference list, data citations must be labeled with "[DATASET]". A data reference must provide the database name, accession number/identifiers and a resolvable link to the landing page from which the data can be accessed at the end of the reference.

12) Author contributions: You will be asked to provide CRediT (Contributor Role Taxonomy) terms in the submission system. These replace a narrative author contribution section in the manuscript.

13) A Conflict of Interest statement should be provided in the main text.

14) Every published paper includes a 'Synopsis' to further enhance discoverability. Synopses are displayed on the journal webpage and are freely accessible to all readers. They include a short stand first (maximum of 300 characters, including space) as well as 2-5 one-sentences bullet points that summarizes the paper. Please write the bullet points to summarize the key NEW findings. They should be designed to be complementary to the abstract - i.e. not repeat the same text. We encourage inclusion of key acronyms and quantitative information (maximum of 30 words / bullet point). Please use the passive voice. Please attach these in a separate file or send them by email, we will incorporate them accordingly.

15) Include a Reagents and Tools Table as part of the Methods section, which can be downloaded from our author guidelines.

***** Reviewer's comments *****

Referee #1 (Remarks for Author):

The authors had adequately addressed the comments.

Referee #2 (Comments on Novelty/Model System for Author):

The revised manuscript demonstrates adequate technical quality, employing appropriate analytical approaches and showing improved methodological transparency compared with the initial submission. Although formal power calculations are not included, the analyses are generally sound and support the authors' conclusions. The study presents a moderate level of novelty by building on established biomarker concepts, while adding incremental value through the integrated analysis of tumor- and blood-based indicators. While the immediate medical impact remains limited, the findings have potential clinical relevance for patient stratification pending further validation in larger, well-designed cohorts. The use of clinically relevant patient samples strengthens the model system; however, some variability in cohort size and design continues to limit the overall strength of the conclusions.

Referee #2 (Remarks for Author):

The authors have clearly engaged with the comments from the previous review round and made a genuine effort to improve the manuscript. The expansion of the validation cohorts, greater consistency in terminology, and inclusion of additional methodological details have all strengthened the technical presentation of the study. In addition, the discussion of potential clinical applications is clearer and more focused than in the original submission. Nevertheless, a key concern remains for a study that aims to propose predictive biomarkers. Although extensive functional validation is not necessarily expected at this stage, the manuscript still lacks a formal power calculation to justify the sizes of the validation cohorts. Different indicators are analyzed using cohorts of varying sizes, and correlations between tumor- and blood-based measurements are presented as supporting evidence. While informative, these analyses cannot substitute for independent validation cohorts whose sizes are determined a priori based on anticipated effect size, statistical power, and confidence level. Without such justification, it is difficult to determine whether the reported associations are robust or reflect cohort-specific variability. Addressing this issue would markedly strengthen both the statistical rigor and the translational credibility of the proposed biomarker framework.

Point-by-point Responses to the Reviewers' Comments**Reviewer #1's Comments (Remarks for Author):**

The authors had adequately addressed the comments.

Response: We sincerely thanked Reviewer #1 for the positive feedback on our revised manuscript.

Reviewer #2 (Comments on Novelty/Model System for Author):

The revised manuscript demonstrates adequate technical quality, employing appropriate analytical approaches and showing improved methodological transparency compared with the initial submission. Although formal power calculations are not included, the analyses are generally sound and support the authors' conclusions. The study presents a moderate level of novelty by building on established biomarker concepts, while adding incremental value through the integrated analysis of tumor- and blood-based indicators. While the immediate medical impact remains limited, the findings have potential clinical relevance for patient stratification pending further validation in larger, well-designed cohorts. The use of clinically relevant patient samples strengthens the model system; however, some variability in cohort size and design continues to limit the overall strength of the conclusions.

Reviewer #2 (Remarks for Author):

The authors have clearly engaged with the comments from the previous review round and made a genuine effort to improve the manuscript. The expansion of the validation cohorts, greater consistency in terminology, and inclusion of additional methodological details have all strengthened the technical presentation of the study. In addition, the discussion of potential clinical applications is clearer and more focused than in the original submission. Nevertheless, a key concern remains for a study that aims to propose predictive biomarkers. Although extensive functional validation is not necessarily expected at this stage, the manuscript still lacks a formal power calculation to justify the sizes of the validation cohorts. Different indicators are analyzed using cohorts of varying sizes, and correlations between tumor- and blood-based measurements are presented as supporting evidence. While informative, these analyses cannot substitute for independent validation cohorts whose sizes are determined a priori based on anticipated effect size, statistical power, and confidence level. Without such justification, it is difficult to determine whether the reported associations are robust or reflect cohort-specific variability. Addressing this issue would markedly strengthen both the statistical rigor and the translational credibility of the proposed

biomarker framework.

Response: We sincerely thank Reviewer #2 for the positive feedback on our revised manuscript regarding technical quality, analytical approaches, and methodological transparency. We also thanked Reviewer #2 for the guidance and comments.

According to the Reviewer's comments, we performed a priori sample-size assessment for the key independent validation cohort used to evaluate candidate tumor-derived indicators for predicting pathologic complete or major responses (pCR/MPR) during research. **The calculation results showed that the minimum number of validation samples was 50, less than our current number of validation cohorts: tumors (n = 54) and blood samples (n = 228).** Moreover, the validation cohort size we calculated was similar to those reported in other studies on response predictors of non-small cell lung cancer (NSCLC) treatment, respectively (Yeong et al, 2021 (n = 35); Kumagai et al, 2020, (n =48); Gettinger et al, 2018, (n=49)). These strengthened the reliability of our size estimation. We also added details about a priori sample-size assessment in the revised Method section (**Lines 870-912 in the Revised Manuscript**).

Lines 870-912 in the Revised Manuscript:

"A priori sample-size assessment for the independent validation cohort

In neoadjuvant lung cancer studies, obtaining high-quality tumor tissue (sufficient material, pre-analytic handling, and assay success) is typically more constrained than obtaining peripheral blood. Accordingly, we prioritized tumor-derived indicator validation as the primary inferential objective and planned the cohort size to achieve adequate statistical power within a logistic-regression framework.

Planned analysis model: The primary endpoint was binary pCR/MPR status ($Y = 1$ for pCR/MPR, $Y = 0$ otherwise). Each candidate tumor indicator was modeled as a continuous predictor and standardized using z-score transformation (mean 0, standard deviation 1), such that the odds ratio (OR) reflects the change in odds of pCR/MPR per 1-standard-deviation (1 – SD) increase in indicator value. The planned single-covariate logistic regression model was:

$$\text{logit}\{\text{Pr}(Y = 1|X)\} = \alpha_0 + \sigma \cdot X, \text{ where } \sigma = \ln(\text{OR}).$$

A priori sample-size calculation: Sample-size planning was performed using the information-based approximation for logistic regression originally developed by Whittemore and summarized in the corrected and tabulated formulation by Hsieh (1989) for a single continuous covariate (Hsieh, 1989). Under this approach, the required total sample size (N) depends on (i) the targeted effect size $\sigma = \ln(\text{OR})$, (ii) the anticipated event probability P at the mean covariate value (i.e., $X = 0$ after standardization), (iii) the two-sided type I error rate α , and (iv) power $(1 - \beta)$. The continuous-covariate correction term described by Hsieh was applied. Design inputs were specified as follows:

- *Target effect size: Pilot analyses suggested a plausible range of $\text{OR} \in [2.0, 2.5]$; we therefore prespecified $\text{OR}_{\text{target}} = 2.3$ as a representative target effect size for study planning.*
- *Type I error and power: two-sided $\alpha = 0.05$ and power $(1 - \beta) = 0.80$.*

- *Event probability: anticipated pCR/MPR rate $P = 0.65$, consistent with published neoadjuvant immunotherapy/chemoimmunotherapy cohorts $P \in [0.6, 0.7]$ and aligned with our regimen and pilot observations (Jiang et al, 2022).*

With these prespecified assumptions, the Hsieh approximation yielded a minimum required validation cohort size of approximately $N \approx 50$ patients.

Achieved independent validation cohort and inferential analyses: Guided by the above a priori rationale and within feasibility constraints, we ultimately assembled an independent tumor validation cohort of 54 patients, with an observed pCR/MPR proportion of 62.9%, meeting the prespecified sample-size requirement. Within this independent cohort, we evaluated two candidate tumor indicators, both analyzed as z-score–standardized continuous variables. Each indicator was assessed using univariable logistic regression, consistent with the single-covariate framework underlying the sample-size calculation. The proportion of classical monocytes among primary tumor monocytes showed an OR of 1.87 (95% CI 1.04–3.39; $p = 0.0379$) per 1 – SD increase, while the ratio of APOBEC3A+ monocytes among total monocytes in native primary tumors showed an OR of 2.29 (95% CI 1.07–4.94; $p = 0.0336$). These effect sizes were statistically significant and consistent with the prespecified design-stage target effect size ($OR_{target} = 2.3$).

In summary, the independent cohort used for tumor indicator validation was supported by an explicit a priori sample-size rationale, and the observed effect sizes at this sample scale were consistent with the prespecified design assumptions, supporting the robustness of the primary validation findings. Moreover, the validation cohort size we calculated is similar to those reported in other studies on response predictors of NSCLC treatment, which strengthened the reliability of our size estimation (Yeong et al, 2021 ($n=35$); Kumagai et al, 2020 ($n=48$); Gettinger et al, 2018 ($n=49$)).”

Additionally, to improve methodological transparency, validation efficiency, and the translational credibility of validation cohorts, we added more detailed clinical information for the validation cohorts, including gender, age, and collection date, in Tables EV4 and EV5. We also added a new Table EV6 to side-by-side show the distribution of clinical information in the peripheral blood and primary tumor validation cohorts to emphasize unbiased sampling. Additionally, to clearly show the temporal independence of the validation cohorts relative to the discovery cohorts, we added Figure 1G (Rebuttal Figure 1) and Figure 2D (Rebuttal Figure 2), which show the discovery and validation cohorts, respectively. Compared to discovery cohorts mostly from 2023, the validation cohorts spanned 2023 to 2024, expanding the temporal range by almost 1 year. It strengthened the validation's efficiency and translational credibility. We also added related descriptions in the revised Result section (**Figure 1G, Lines 99-103 in the Revised Manuscript; Figure 2D, Lines 142-147 in the Revised Manuscript**).

Lines 99-103 in the Revised Manuscript:

“To validate the positive relationship between the native blood monocyte ratio and the responsiveness of LUSC patients, we collected complete blood count (CBC) test data from LUSC patients before treatment ($n=228$, Table EV4). Compared to the discovery cohort mainly from

2023 (Table EV1), we collected more native CBC data of patients from 2024 to strengthen the translational credibility (Fig. 1G; Table EV4 and EV6).”

Lines 142-147 in the Revised Manuscript:

“To validate this result, we collected pre-treatment primary tumor samples for IF staining (Table EV5). Given that the discovery cohort was primarily from 2023 (Table EV1), we augmented it with native tumor samples collected in 2024 to strengthen the translational robustness of our results (Fig. 2D; Table EV6). We also performed a priori sample-size assessment for the key independent tumor-tissue validation cohort used to evaluate candidate tumor-derived indicators for predicting pCR/MPR (Detailed in Methods).”

Figure 1G ; Rebuttal Figure 1. Collection time distribution of the peripheral blood discovery and validation cohorts.

Figure 2D ; Rebuttal Figure 2. Collection time distribution of the primary tumor discovery and validation cohorts.

Partial Table EV4. The Clinical Information of CBC Test Cohort

Sampling ID	Collection Date (Y/M/D)	0-pCR;1-MPR;2-non-MPR	Gender 1-Male 0-Female	Age	ological T	White Blood Cell Count (10 ⁹ /L)	Absolute Neutrophil Count (10 ⁹ /L)	Absolute Lymphocyte Count (10 ⁹ /L)	Absolute Monocyte Count (10 ⁹ /L)	Red Blood Cell Count (10 ¹² /L)	Hemoglobin (g/L)	Platelet Count (10 ⁹ /L)
id_001	2022/11/30	2	1	67	LUSC	5.4	3.88	1.23	0.24	4.83	146	310
id_002	2023/3/25	0	1	62	LUSC	8.4	6.23	1.56	0.36	4.05	136	215
id_003	2023/4/28	2	1	72	LUSC	6	3.66	1.84	0.32	3.18	97	245
id_004	2023/5/20	0	1	72	LUSC	8.6	6.51	1.18	0.41	4.4	118	332
id_005	2023/4/7	2	1	62	LUSC	9.9	6.54	2.25	0.65	4.78	144	289
id_006	2023/8/9	2	1	68	LUSC	6.5	4.38	1.5	0.35	4.14	145	354
id_007	2023/8/27	0	1	72	LUSC	6.1	3.97	1.15	0.72	4.38	127	374
id_008	2023/8/26	2	1	63	LUSC	7.7	4.63	1.64	0.37	4.3	126	312
id_009	2023/9/4	2	1	67	LUSC	8.6	6.32	1.96	0.27	4.62	151	180
id_010	2023/9/12	1	1	53	LUSC	7.2	4.61	2.04	0.45	4.98	152	179
id_011	2023/10/8	0	1	60	LUSC	7	4.73	1.67	0.41	4.91	149	200
id_012	2024/3/23	1	1	75	LUSC	5.5	3.56	1.39	0.26	4.31	134	224
id_013	2024/3/11	1	1	69	LUSC	3.8	2.66	0.84	0.23	4.49	143	243
id_014	2023/5/25	2	1	69	LUSC	6.2	3.83	1.88	0.3	4.46	142	155
id_015	2024/2/7	0	1	68	LUSC	8.8	5.81	2.31	0.57	5.35	116	203
id_016	2024/1/3	0	1	57	LUSC	13.2	8.45	4.05	0.65	3.18	105	219
id_017	2024/1/31	0	1	72	LUSC	9.8	7.15	1.72	0.74	4.12	123	312
id_018	2024/2/15	1	1	62	LUSC	4.8	3.5	0.89	0.3	4.16	126	177
id_019	2024/1/16	1	1	64	LUSC	8.1	5.61	1.63	0.54	4.68	136	399
id_020	2024/1/6	0	1	53	LUSC	7.4	4.63	1.99	0.46	4.01	132	252
id_021	2023/11/22	0	1	64	LUSC	9.1	6.85	1.29	0.72	3.35	100	456
id_022	2023/8/18	2	1	65	LUSC	3.8	2.31	1.27	0.19	4.86	151	243
id_023	2024/7/15	1	1	67	LUSC	6	4.46	0.98	0.41	4.73	137	244
id_024	2024/7/2	2	1	69	LUSC	7.6	2.1	4.45	0.37	4.86	147	139
id_025	2024/5/29	2	1	67	LUSC	5.3	2.31	1.65	0.3	4.1	128	161
id_026	2024/5/27	2	1	74	LUSC	5.8	3.25	1.9	0.34	4.26	142	212
id_027	2024/7/23	0	1	53	LUSC	6.3	4.44	1.32	0.42	4.34	139	233
id_028	2024/6/6	0	1	73	LUSC	5.3	3.13	1.51	0.49	4.25	114	131
id_029	2024/6/2	0	1	58	LUSC	6.2	3.92	1.46	0.49	4.07	121	304
id_030	2024/6/13	0	1	69	LUSC	11.9	10.2	1.15	0.5	3.59	99	358
id_031	2024/3/5	0	1	67	LUSC	5.6	3.11	2.09	0.22	4.77	150	262
id_032	2024/1/11	0	1	68	LUSC	10	6.62	2.43	0.82	4.64	139	420
id_033	2024/3/29	0	1	67	LUSC	6.3	4.54	1	0.55	3.86	104	200
id_034	2024/8/26	0	1	67	LUSC	8.7	5.95	1.71	0.67	5.38	151	300
id_035	2023/2/2	0	1	68	LUSC	6.6	4.6	1.38	0.47	4.94	152	281
id_036	2023/3/7	0	1	66	LUSC	7.9	5.2	2.06	0.49	4.3	129	192
id_037	2023/3/25	1	1	62	LUSC	8.4	6.23	1.56	0.36	4.05	136	215
id_038	2023/4/7	2	1	62	LUSC	9.9	6.54	2.25	0.65	4.78	144	289
id_039	2023/3/4	0	1	60	LUSC	8.6	6.68	1.01	0.83	4.06	125	430
id_040	2023/4/19	1	1	56	LUSC	7.4	4.69	1.93	0.32	4.71	145	296
id_041	2023/1/9	2	1	43	LUSC	6.9	4.91	1.55	0.27	4.97	151	309
id_042	2023/1/26	2	1	53	LUSC	6.6	4.49	1.51	0.43	5.41	148	265
id_043	2023/2/14	2	1	59	LUSC	7.1	4.39	2.18	0.44	4.48	145	284
id_044	2023/2/23	0	1	69	LUSC	7.3	4.74	1.9	0.35	4.86	152	267
id_045	2023/2/16	2	1	68	LUSC	7.3	4.55	2.07	0.45	4.54	134	218
id_046	2024/9/12	0	1	73	LUSC	5.3	3.61	1.24	0.32	4.61	138	192
id_047	2024/9/24	1	1	69	LUSC	7.7	4.35	2.72	0.39	5.27	166	243
id_048	2024/8/24	0	1	49	LUSC	10.2	7.04	2.1	0.95	4.57	139	345
id_049	2024/8/19	1	1	73	LUSC	6.8	4.82	1.31	0.54	3.91	124	264
id_050	2024/9/2	2	1	75	LUSC	6.6	5.14	0.67	0.5	4.2	129	252
id_051	2024/8/17	1	1	46	LUSC	10.2	7.39	2.01	0.51	4.37	147	327
id_052	2024/7/3	0	1	76	LUSC	6	4.05	1.3	0.43	4.25	125	188
id_053	2024/8/18	1	1	62	LUSC	6.9	4.91	1.35	0.33	5.56	136	230
id_054	2024/9/8	0	1	72	LUSC	7.5	4.78	2.06	0.54	4.59	144	214
id_055	2024/8/26	0	1	67	LUSC	8.7	5.95	1.71	0.67	5.38	151	300
id_056	2024/7/30	0	1	59	LUSC	10.3	8.14	1.52	0.53	4.35	136	474
id_057	2024/7/17	2	1	61	LUSC	7.8	5.65	1.54	0.45	4.87	135	276
id_058	2024/7/31	0	1	72	LUSC	11.1	9.24	1.04	0.71	4.14	120	305
id_059	2024/7/30	2	1	68	LUSC	5.5	3.73	1.19	0.29	5	152	258
id_060	2024/7/19	2	1	65	LUSC	10.5	8.54	1.26	0.62	3.58	100	299
id_061	2024/7/20	2	0	58	LUSC	7.8	5.95	1.01	0.48	3.89	116	175
id_062	2024/6/25	1	1	72	LUSC	9.8	7.44	1.84	0.41	4.43	139	278
id_063	2024/8/23	0	1	74	LUSC	8.7	6.95	0.91	0.72	3.89	121	180
id_064	2024/7/17	0	1	65	LUSC	6.3	3.44	2.15	0.44	4.9	149	289
id_065	2024/7/3	0	1	63	LUSC	6.8	4.13	1.71	0.57	3.6	121	267
id_066	2024/5/28	0	1	54	LUSC	9	7.42	1	0.5	5.16	148	118
id_067	2024/8/8	0	1	75	LUSC	6.8	4.32	1.87	0.37	4.24	139	178
id_068	2024/6/17	0	1	70	LUSC	9.3	7.49	1.3	0.43	3.43	95	315
id_069	2024/7/31	1	1	60	LUSC	12.4	9.63	2.07	0.66	5.05	149	231
id_070	2024/7/31	0	1	72	LUSC	6.6	4.45	1.49	0.55	4.04	134	168

Table EV5. The Clinical Information of IF Staining Cohort

sample_ID	Residual Viable Tumor	Gender	Age	Pathological Type	Collection Date (Y.M.D)
IF_01	0%	Male	66	LUSC	2023.10.10
IF_02	0%	Male	64	LUSC	2023.10.23
IF_03	0%	Male	72	LUSC	2023.10.23
IF_04	0%	Male	60	LUSC	2023.09.22
IF_05	0%	Male	71	LUSC	2023.07.30
IF_06	0%	Male	62	LUSC	2023.05.26
IF_07	0%	Male	60	LUSC	2023.11.27
IF_08	0%	Male	68	LUSC	2024.04.30
IF_09	0%	Male	72	LUSC	2024.04.17
IF_10	0%	Male	58	LUSC	2024.08.14
IF_11	0%	Male	53	LUSC	2024.08.07
IF_12	0%	Male	53	LUSC	2024.02.26
IF_13	5%	Male	75	LUSC	2024.05.30
IF_14	5%	Male	62	LUSC	2024.04.15
IF_15	8%	Male	69	LUSC	2024.05.10
IF_16	13%	Male	69	LUSC	2023.11.13
IF_17	15%	Male	69	LUSC	2024.05.08
IF_18	20%	Male	67	LUSC	2024.09.04
IF_19	30%	Male	74	LUSC	2024.08.19
IF_20	50%	Male	69	LUSC	2024.09.06
IF_21	60%	Male	65	LUSC	2023.10.10
IF_22	60%	Male	62	LUSC	2023.06.05
IF_23	60%	Male	72	LUSC	2023.06.26
IF_24	95%	Male	66	LUSC	2023.03.07
IF_25	0%	Male	77	LUSC	2024.09.09
IF_26	0%	Male	66	LUSC	2024.03.27
IF_27	0%	Male	61	LUSC	2024.10.11
IF_28	0%	Male	75	LUSC	2024.05.31
IF_29	0%	Male	64	LUSC	2024.08.31
IF_30	0%	Male	70	LUSC	2024.09.27
IF_31	0%	Male	66	LUSC	2024.05.08
IF_32	0%	Male	67	LUSC	2024.03.08
IF_33	0%	Male	67	LUSC	2024.10.24
IF_34	1%	Male	72	LUSC	2024.05.08
IF_35	2%	Male	69	LUSC	2024.11.21
IF_36	2%	Male	56	LUSC	2024.06.07
IF_37	5%	Male	62	LUSC	2024.10.30
IF_38	5%	Male	75	LUSC	2024.11.05
IF_39	5%	Male	66	LUSC	2024.03.29
IF_40	5%	Male	73	LUSC	2024.06.12

IF_41	8%	Male	60	LUSC	2024.09.26
IF_42	<10%	Male	76	LUSC	2024.07.09
IF_43	10%	Male	73	LUSC	2024.11.13
IF_44	20%	Male	67	LUSC	2024.05.14
IF_45	30%	Male	65	LUSC	2024.10.15
IF_46	40%	Male	66	LUSC	2024.09.05
IF_47	50%	Male	78	LUSC	2024.09.19
IF_48	60%	Male	70	LUSC	2024.11.14
IF_49	70%	Male	76	LUSC	2024.03.12
IF_50	70%	Male	78	LUSC	2024.04.25
IF_51	70%	Male	64	LUSC	2024.11.29
IF_52	70%	Male	63	LUSC	2024.07.27
IF_53	80%	Male	65	LUSC	2024.11.20
IF_54	>90%	Male	58	LUSC	2024.06.26

Table EV6. The Clinical Characteristic Distribution Table of Validation Cohort

Characteristic	Patients of Complete Blood Count (N=228)	Patients of Tumor Slides (N=54)
Sampling Year, n (%)		
2022	11 (4.8)	0(0)
2023	111(48.7)	12(22.2)
2024	106 (46.5)	42 (77.8)
Age, years		
Median (Range)	66 (43-82)	67(53-78)
Gender, n (%)		
Male	224(98.2)	54(100.0)
Female	4(1.8)	0(0)
Pathological Type, n (%)		
Lung Squamous Cell Carcinoma (LUSC)	228 (100.0)	54 (100.0)
Response Status, n (%)		
Pathologic Complete or Major Responses	181(79.4)	33(61.1)
Non-Major Pathological Response	47(20.6)	21(38.9)

17th Feb 2026

Dear Prof. Wang,

Thank you for submitting your revised study. We have now received the report from referee #2, who is satisfied with the revisions. I will therefore be able to accept your manuscript once the following editorial matters are addressed:

1/ Manuscript text:

- Please remove the yellow highlights and indicate in track changes mode any new modification.
- Please note that all corresponding authors are required to supply an ORCID ID for their name upon submission of a revised manuscript. An ORCID identifier is missing for Prof. Kai Wang.
- Please correct the headings and order of the manuscript sections to: Abstract / The Paper Explained / Introduction / Results / Discussion / Methods / Data Availability / Acknowledgements / Disclosure and Competing Interests Statement / References / Figure Legends / Tables / Expanded View Figure Legends.
- Remove "resource availability" headings, as well as "lead contact", and "materials availability" sections.
- Remove the "Supplementary information" list with the appendix and table legends.
- Methods:
 - o Remove the Reagents and Tools table from the manuscript text.
 - o "Ethics Declaration" should be moved to the Methods.
 - o Human samples: please include a statement confirming that the experiments conformed to the principles set out in the WMA Declaration of Helsinki and the Department of Health and Human Services Belmont Report.
 - o Biorender: remove from the reagent and tools table and add a dedicated section to the Methods using this format:
Graphics:
(some of the... OR Figure #... OR synopsis) Graphics were created with BioRender.com.
- Acknowledgements: please note that the funding information provided in the manuscript should match the submission system. Currently, LR22H160002 is only partially displayed in the submission system.
- Author contribution: please remove from the manuscript text and provide instead CRediT (Contributor Role Taxonomy) terms in the submission system.
- Declaration of interest should be renamed "Disclosure of Competing Interest Statement".

2/ Figures:

- Main figures and EV figures should be uploaded as separate, high resolution figure files.
- The legends of the EV Tables should be added directly to the tables and the tables should be uploaded as individual files.
- Appendix: please move the figure legends to the corresponding figures so that the readers can see them displayed together.
- Callouts: all figure and figures panels should be referenced in the text, and in sequential order. Currently, a callout is missing for Appendix Figure 3, and Table EV4 is called out after Table EV2, Table EV5 after Table EV6, etc.
- Please provide a legend for Figure 4.
- Please address the queries from our data editors in the figure legends:
 1. Please note that the exact p values are not provided in the legends of figures EV1 D
 2. Please indicate the statistical test used for data analysis in the legends of figures 3A, EV3A, EV4 B, EV4 D, S1 B, S2 A, S3 A
 3. Please note that the box plots need to be defined in terms of minima, maxima, centre, bounds of box and whiskers, and percentile in the legends of figures 1F, H; 2F, 3D, E; EV1 A, D, F, G; EV2 C, S1 A, S2 A, S3 A
 4. Please note that information related to n is missing in the legends of figures 1F, 3A, 3D, EV1 A, D, F, G; EV2 C, EV3 A, EV4 B, D; S1 A, B; S2 A, S3 A
 5. Please note that the error bars are not defined in the legend of figure EV4 D.

3/ Source Data: Please provide minimally processed staining data for Figure 2E and Figure 3E. Source data files should be uploaded as one (zipped) files per figure.

4/ Synopsis image: please provide a simpler visual abstract, different from Figure 4, as a jpeg/tiff/PNG file 550 px wide x 300-600 px high.

5/ As part of the EMBO Publications transparent editorial process initiative (see our Editorial at <http://embomolmed.embopress.org/content/2/9/329>), EMBO Molecular Medicine will publish online a Review Process File (RPF) to accompany accepted manuscripts.

This file will be published in conjunction with your paper and will include the anonymous referee reports, your point-by-point response and all pertinent correspondence relating to the manuscript. Let us know whether you agree with the publication of the RPF and as here, if you want to remove or not any figures from it prior to publication.

I look forward to receiving your revised manuscript.

Yours sincerely,

Lise Roth

***** Reviewer's comments *****

Referee #2 (Remarks for Author):

The authors have satisfactorily addressed my
questions.

The authors addressed the remaining editorial issues.

9th Mar 2026

Dear Prof. Wang,

Thank you for submitting your revised files. I am pleased to inform you that your manuscript is accepted for publication and is now being sent to our publisher to be included in the next available issue of EMBO Molecular Medicine.

You may qualify for financial assistance for your publication charges - either via a Springer Nature fully open access agreement or an EMBO initiative. Check your eligibility: <https://link.springer.com/journal/44321/how-to-publish-with-us>

Yours sincerely,

Lise Roth

>>> Please note that it is EMBO Molecular Medicine policy for the transcript of the editorial process (containing referee reports and your response letter) to be published as an online supplement to each paper. If you do NOT want this, you will need to inform the Editorial Office via email immediately. More information is available here: <https://link.springer.com/partners/embo-press/editorial-policies#Peer%20review>